# Accounting for interactions between Sustainable Development Goals is essential for water pollution control in China

Mengru Wang [1,2✉], Annette B. G. Janssen [2], Jeanne Bazin[2], Maryna Strokal [2], Lin Ma [1✉] & Carolien Kroeze[2]

Meeting the United Nations' (UN's) 17 Sustainable Development Goals (SDGs) has become a worldwide mission. How these SDGs interrelate, however, is not well known. We assess the interactions between SDGs for the case of water pollution by nutrients in China. The results show 319 interactions between SDGs for clean water (SDGs 6 and 14) and other SDGs, of which 286 are positive (synergies) and 33 are negative (tradeoffs) interactions. We analyze six scenarios in China accounting for the cobenefits of water pollution control using a large-scale water quality model. We consider scenarios that benefit from synergies and avoid tradeoffs. Our results show that effective pollution control requires accounting for the interactions between SDGs. For instance, combining improved nutrient management, efficient food consumption, and climate mitigation is effective for simultaneously meeting SDGs 6 and 14 as well as other SDGs for food, cities and climate. Our study serves as an example of assessing SDG interactions in environmental policies in China as well as in other regions of the world.

[1] Key Laboratory of Agricultural Water Resources, Center for Agricultural Resources Research, Institute of Genetics and Developmental Biology, Chinese Academy of Sciences, 286 Huaizhong Road, Shijiazhuang 050021, China. [2] Water Systems and Global Change Group, Wageningen University & Research, Droevendaalsesteeg 3, 6708 PB Wageningen, the Netherlands. ✉email: mengru.wang@wur.nl; malin1979@sjziam.ac.cn

The United Nations (UN) introduced 17 Sustainable Development Goals (SDGs) to achieve a better and more sustainable future for all[1], and thus, meeting these goals has become a worldwide mission. Although SDGs are formulated as individual goals, there are significant interactions between them, which lead to synergies and tradeoffs[2–5]. In other words, meeting one SDG may positively or negatively (via synergies or tradeoffs, respectively) affect another SDG. Taking "Clean Water and Sanitation" (SDG 6) as an example, there are synergies and tradeoffs with several other SDGs, such as SDG 2 "Zero Hunger". A synergy occurs when increased water use efficiency for food production (Target 2.4) also reduces water scarcity (Target 6.4) and ensures a water supply for all. Tradeoffs occur when, for example, water demand for sanitation and hygiene systems (Target 6.2) competes with that for agriculture (Target 2.3). Currently, there is a limited understanding of SDG interactions[6]. Ignoring these synergies and tradeoffs may lead to the enactment of policies that are ineffective for meeting these SDGs.

The increasing amount of nutrients in rivers has deteriorated water quality worldwide[7–9]. This fact also holds for China, posing threats to both society and ecosystems[10,11]. Between 2000 and 2012, 28% of groundwater samples in China exceeded the World Health Organization's (WHO's) maximum contaminant level ($10\,mg\,N\,L^{-1}$)[12]. In 2010, up to 36% of river sections and 40% of major lakes in China did not meet the quality criteria to be drinking water sources[13]. Moreover, excessive nutrients in rivers are transported to seas, which caused approximately 500 episodes of harmful algae blooms in China's coastal seas from 2006 to 2012, leading to adverse effects on human health and coastal ecosystems[14]. This situation is expected to worsen in the future because of socioeconomic developments and climate change[7,15,16].

Reducing nutrient pollution in rivers and coastal waters is thus important for China, especially given its commitment to the SDGs under the UN's Agenda 2030[17,18]. For clean water bodies, SDG 6 "Clean Water and Sanitation" and SDG 14 "Life Below Water" are particularly important. However, Stiftung[19] showed that with the current policies, China is behind schedule in terms of meeting most of the SDGs by 2030. Fast population growth, slow progress toward sustainable production and consumption, and the negative impacts of the ongoing coronavirus disease 2019 (COVID-19) pandemic on socioeconomic development make it difficult for China to meet the SDGs[20]. Therefore, effective water pollution control in China is urgently needed.

Here, we analyze the interactions (synergies and tradeoffs) between the goals and targets of the SDGs for pollution control, exemplifying the case of water pollution by nutrients in China. These interactions have been identified and assessed using a seven-point-scale framework (see "Methods" and Supplementary Table 2 for a description of the framework) based on the existing literature (Supplementary Table 1) and our expert knowledge about water pollution in China. Next, we develop and explore future scenarios promoting the synergies and mitigating the tradeoffs between the selected SDGs, with a focus on agriculture, sewage, food consumption, and climate mitigation. Our analysis thus supports water pollution control as well as policies to achieve sustainable development and ensure a better future for all in China as well as in other regions of the world.

## Results

### Fifty-one targets of the SDGs are relevant to nutrient pollution. All 17 SDGs have targets that are relevant to nutrient pollution in rivers and coastal waters in China. We identify 51 targets that we consider to be of low, moderate, or high relevance for nutrient pollution in water (Fig. 1 and Supplementary Table 1). Of these, 13 targets are considered of high relevance,

addressing the direct sources of nutrient losses to water systems. We consider 9 targets to be of moderate relevance, addressing the impacts of nutrient pollution on aquatic ecosystems and human health or influencing the resilience of ecosystems to nutrient pollution. Finally, 29 targets are of low relevance, addressing technological, social, administrative or economic interventions indirectly related to nutrient pollution in water systems.

Two SDGs are particularly important for nutrient pollution in water: SDG 6 "Clean Water and Sanitation" and SDG 14 "Life Below Water". Most of the targets for achieving SDGs 6 and 14 call for nutrient pollution control. All SDG 6 targets are relevant to this, and of which 4, 1, and 3 targets have high, moderate, and low relevance, respectively. SDG 14 has 8 relevant targets, of which 1 target has high relevance, 3 have moderate relevance, and 4 have low relevance (Fig. 1).

Other SDGs that are strongly related to nutrient pollution are SDG 2 "Zero Hunger", SDG 11 "Sustainable Cities and Communities", SDG 12 "Responsible Consumption and Production" and SDG 13 "Climate Action". These SDGs have a total of 12 relevant targets, of which 8 are of high relevance because they affect the human activities that are responsible for diffuse and point sources of nutrient pollution (Fig. 1). The remaining SDGs have only targets of moderate or low relevance for nutrient pollution in waters. These SDGs are relevant either through the impacts of water pollution by nutrients on nature (e.g., SDG 15 "Life on Land") and society (e.g., SDG 1 "No Poverty" and SDG 3 "Good Health & Well-being") or through contributing to managing nutrient pollution through education (e.g., SDG 4 "Quality Education"), financial support (e.g., SDG 8 "Decent Work and Economic Growth"), and collaboration among sectors and countries (e.g., SDG 17 "Partnerships").

### Three hundred nineteen synergies and tradeoffs between SDGs. We identify 319 interactions between SDGs 6 and 14 and other SDGs, of which 286 are positive (synergies) and 33 are negative (tradeoffs) (see Figs. 2 and 3 and Supplementary Tables 3, 4 for explanations of each of the interactions). These interactions refer to relations between SDG targets that are relevant to nutrient pollution in water systems.

Important interactions are identified between SDGs for sustainable water management (SDGs 6 and 14) and those addressing the drivers of nutrient pollution in water (i.e., agriculture, urbanization, food consumption, and climate action (SDGs 2, 11, 12, and 13, respectively)). We find 82 synergies and 22 tradeoffs between SDGs 6 and 14 and SDGs 2, 11, 12, and 13. Potential tradeoffs can occur mainly between SDG 6 (or SDG 14) and those SDGs that aim to achieve food security with increasing agriculture (SDG 2) and expand housing and basic services in cities and communities (SDG 11). These tradeoffs can be counterbalanced by synergies, particularly through support for SDG 12, which fosters efforts toward increasing resource-use efficiency in all sectors. For example, SDG 12 involves actions aimed at sustainable consumption and production, the efficient use of natural resources and halving food waste. All these factors contribute to reducing nutrient pollution in waters from agriculture and cities and thus to SDGs 6 and 14.

Interactions are also identified between SDGs for sustainable water management (SDGs 6 and 14) and the remaining SDGs (those other than SDGs 2, 11, 12, and 13). These interactions include 194 synergies and 10 tradeoffs (Figs. 2 and 3). Several SDGs call for social (SDG 4, 5, and 10), financial (SDG 8), technical (SDGs 7 and 9) and institutional (SDG 16 and 17) support for sustainable lifestyles that enable society to reduce and control water pollution. Reducing water pollution may benefit poor individuals by reducing their exposure to environmental pollution (SDG 1),

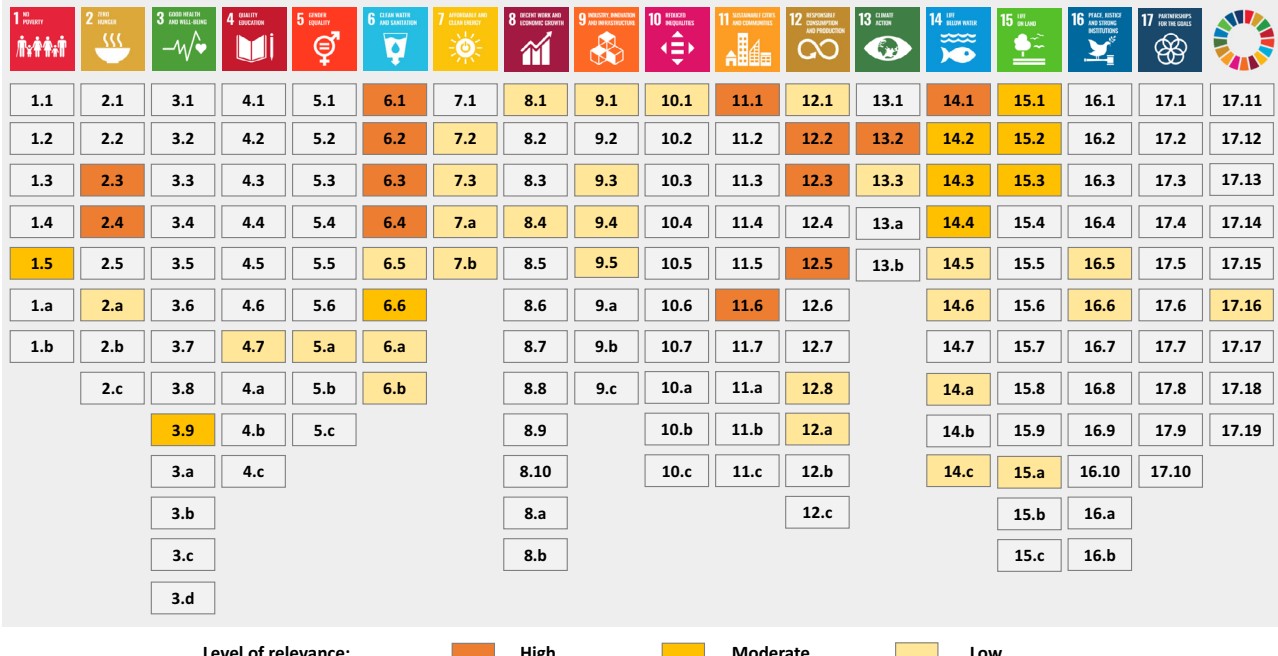

**Fig. 1 Sustainable Development Goals (SDGs) and their targets that are relevant to nutrient pollution in water systems in China.** Three levels of the relevance of targets to nutrient pollution in water systems are considered: "high", "moderate" and "low". Targets classified as having "high" relevance address the direct sources of nutrient losses to the Chinese water systems. Targets classified as having "moderate" relevance address the impacts of nutrient pollution on aquatic ecosystems and human health or influence the resilience of ecosystems to nutrient pollution. Targets classified as having "low" relevance refer to technological, social, administrative or economic interventions indirectly related to nutrient pollution in water systems. The approach to assessing the relevance of the SDGs is described in the "Methods" section. The relevance of these SDGs is explained in Supplementary Table 1.

improving their health and well-being (SDG 3), and allowing for healthier ecosystems on land (SDG 15). Conversely, SDGs 1, 7, 9, and 15 can negatively affect SDGs 6 and 14 by causing water pollution indirectly through the atmospheric deposition of air pollutants from energy use (SDG 7) or industry (SDG 9). In addition, SDGs 1 and 15 may be negatively affected by the side effects of the targets of SDGs 6 and 14 on people's income (SDG 1) or terrestrial ecosystems (SDG 15). For example, management options that limit agricultural production to improve water quality (SDGs 6 and 14) may lower the income of farmers, making it more difficult to achieve SDG 1. Reducing nutrient pollution in rivers (SDG 6) by recycling animal manure on land may lead to higher ammonia ($NH_3$) emissions from agriculture, resulting in higher atmospheric nitrogen (N) deposition in terrestrial ecosystems (SDG 15).

**Future scenarios for China accounting for the cobenefits of water pollution control.** To illustrate the cobenefits of water pollution control, we develop six future scenarios for China, which aim at reducing water pollution while promoting the synergies and mitigating the tradeoffs between relevant SDGs (Fig. 4; "Methods"). We focus on SDGs 6 and 14 and four SDGs (2, 11, 12, and 13) that are highly relevant to the drivers of nutrient pollution (agriculture, sewage, food consumption, and climate mitigation). We explore how pollution control measures, as assumed in these scenarios, may reduce nutrient pollution and discuss the potential cobenefits of other SDGs in China. We quantify total dissolved N (TDN) and total dissolved phosphorous (TDP) inputs to six large rivers and river export of TDN and TDP to seas in 2012 and 2050 using the Model to Assess River Inputs of Nutrients to seAs (MARINA) 2.0 model[16]. Moreover, 2050 is selected because studies indicate that meeting all SDGs by 2030 may be too ambitious[19,21,22], especially given the impacts of

the COVID-19 pandemic on socioeconomic developments[20]. We analyze the scenarios with respect to SDGs 6 and 14 and with a focus on nutrient pollution in rivers and coastal waters. For 2050, we adopt the Shared Socioeconomic Pathway 5-Representative Concentration Pathway 8.5 (SSP5-RCP8.5) from Wang, et al.[16] as the baseline scenario, as it reflects a future with relatively high global warming and conventional nutrient management practices. Based on the SSP5-RCP8.5, five alternative scenarios are developed, named SE (improved sewage treatment), AG (improved nutrient use efficiencies in agriculture), AG + SE (combination of AG and SE), AG + SE + SFC (sustainable food consumption in addition to AG + SE), and AG + SE + SFC + CLI (climate mitigation in addition to AG + SE + SFC) scenarios.

Our scenario analysis shows that accounting for interactions between SDGs (2, 11, 12, and 13) is important to effectively reduce water pollution in China (see Fig. 5 and Supplementary Figs. 9–12 for the results of each scenario). Nutrient pollution levels in six large rivers and coastal waters are projected to be relatively high in the baseline scenarios (the SSP5-RCP8.5). In 2050, inputs to rivers are 18.6 Tg for TDN and 1.3 Tg for TDP (Fig. 5). These nutrients are transported by rivers, leading to the export of 4.3 Tg of TDN and 0.4 Tg of TDP to Chinese seas. TDN and TDP in rivers and coastal waters mainly originate from human activities on land, including direct discharges of animal manure, use of synthetic fertilizers, and discharge of (treated or untreated) human waste (Fig. 5 and Supplementary Figs. 9–12). Water pollution by nutrients is reduced in the alternative scenarios. The lowest levels are calculated for AG + SE + SFC + CLI (Fig. 5 and Supplementary Figs. 9–12). In the AG + SE + SFC + CLI scenario, TDN and TDP inputs to rivers are 64% and 90% lower than in the baseline, respectively. As a result, the river export of TDN is 68% lower and that of TDP is 91% lower than in the baseline scenario. Pollution control measures reducing nutrients contribute to meeting SDGs 6 and 14 as well as other

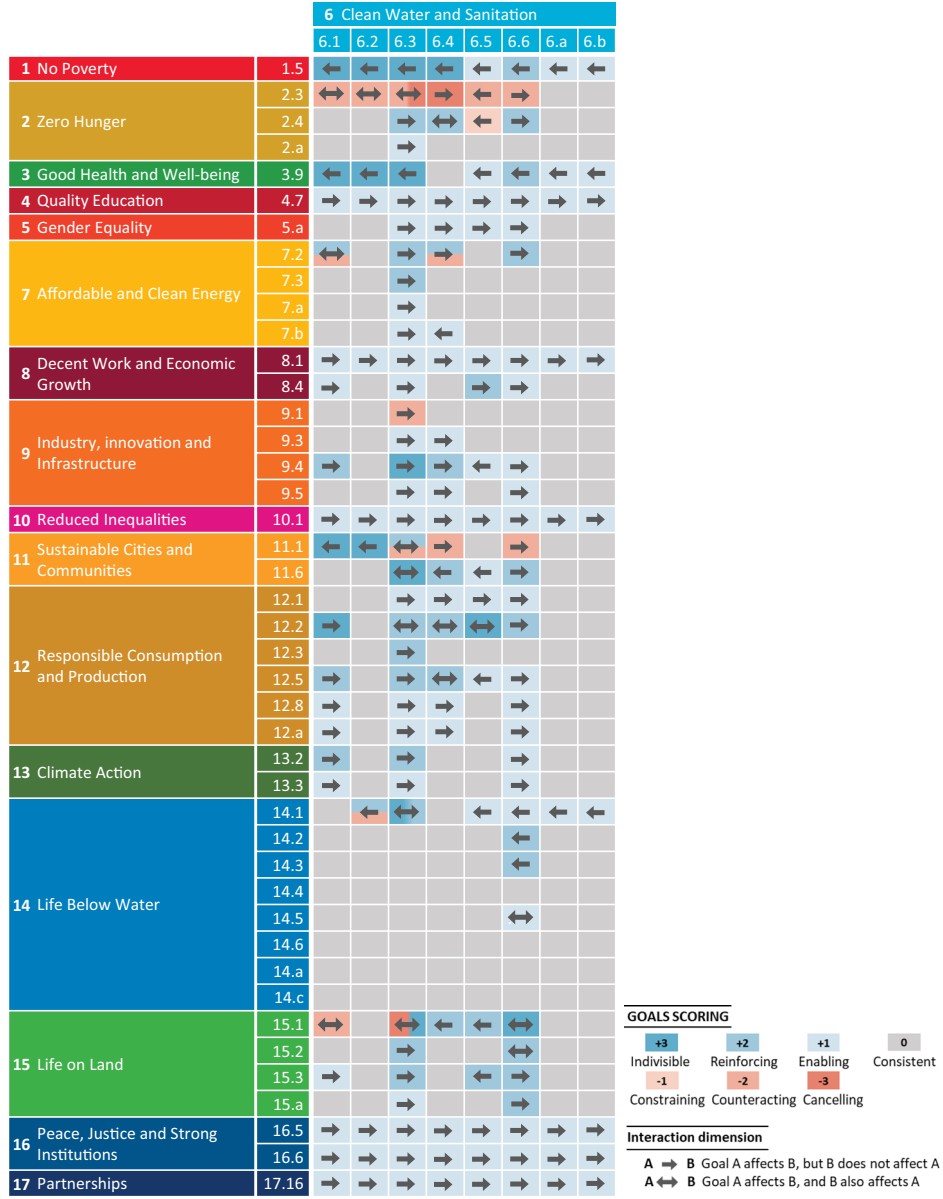

**Fig. 2 Interactions between SDG 6 and other SDGs at the target level in China.** SDGs are Sustainable Development Goals introduced by the United Nations[1]. Here, we focus on the interactions between the targets that are considered relevant to nutrient pollution in water systems (Fig. 1). The approach to assessing the interactions between these SDGs is described in the "Methods" section. The interactions are explained in Supplementary Tables 3, 4.

SDGs that cobenefit from these interventions. Below, we discuss the contributions of these measures to the SDGs.

**Future water pollution control for SDG 6.** We calculate that the water quality at 23 subbasin outlets of 6 large Chinese rivers is worse than China's water quality standard "grade III"[23] (a standard for good ambient water to assess SDG 6) for either dissolved inorganic N (DIN) (1.0 mg/L) or TDP (0.2 mg/L) in 2012 (Figs. 6, 7). The levels of nutrient pollution in rivers remain high in 2050 in the baseline SSP-RCP8.5 scenario due to high nutrient inputs to rivers (Figs. 6, 7). The alternative scenarios may meet SDG 6 by reducing nutrient pollution in rivers in the future (Figs. 6, 7). The effectiveness varies among scenarios and subbasins. In the most effective scenario, $AG + SE + SFC + CLI$, river water quality is improved considerably compared to the baseline scenario (SSP5-RCP8.5) (Figs. 6, 7). We estimate water quality at all subbasin

outlets better than grade III based on TDP concentrations. Only the outlets of 7 subbasins may have DIN concentrations that meet the standard for "grade III", showing that one of the challenges to meeting SDG 6 for China is to reduce N pollution in rivers, which is in line with existing studies[11,24] that concluded that reducing N pollution for a clean water environment in China is challenging because of the additional sources of N loss (e.g., atmospheric N deposition and biological $N_2$ fixation) and that P is more easily removed than N from wastewater treatment.

**Future water pollution control for SDG 14.** We calculate high potentials for coastal eutrophication at the river mouths of all six rivers in 2012 and 2050 for the SSP5-RCP8.5, indicated by positive indicator for coastal eutrophication potential (ICEP) values (Fig. 8). For example, ICEP values range from 2 to 45 C-eq. $km^{-2}$ $year^{-1}$ for the six rivers in 2050 for the SSP5-RCP8.5.

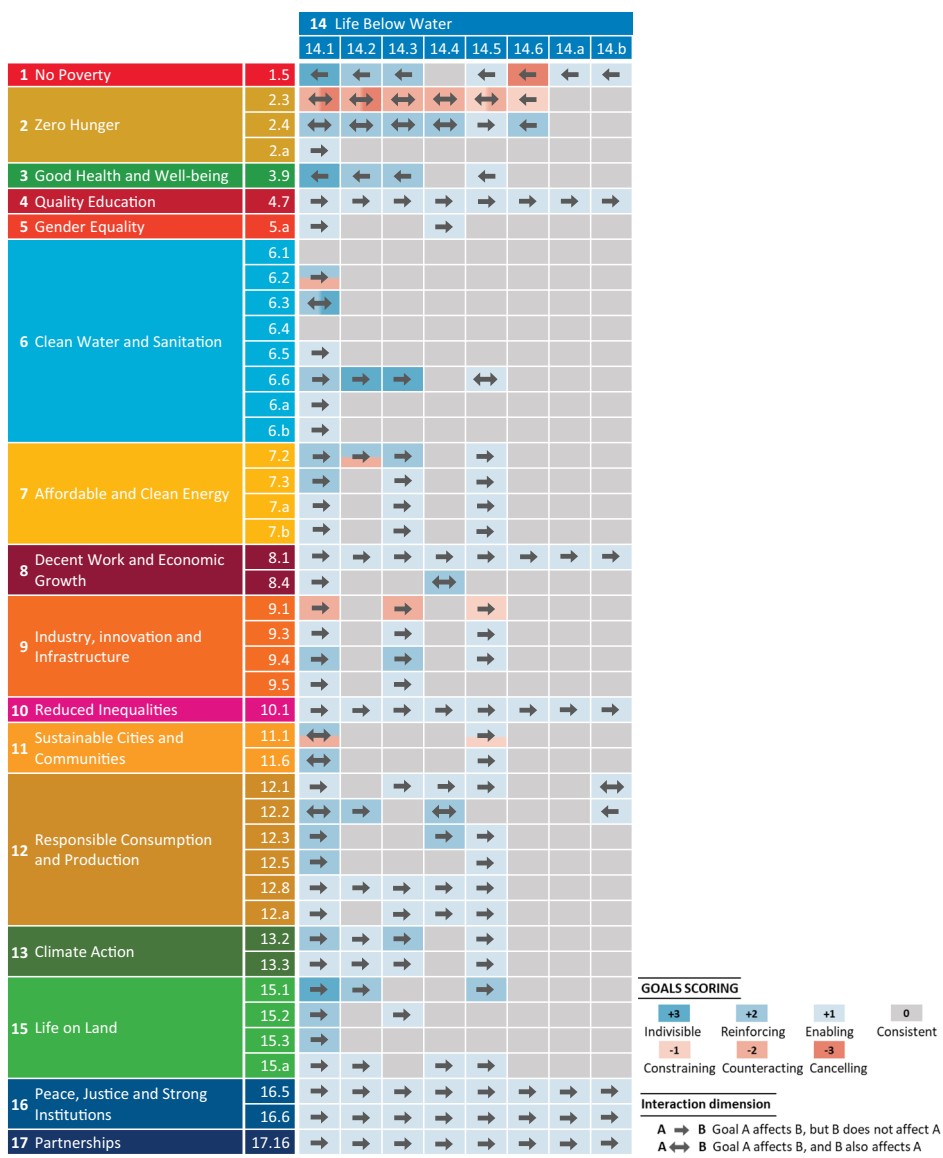

**Fig. 3 Interactions between SDG 14 and other SDGs at the target level in China.** SDGs are Sustainable Development Goals introduced by the United Nations[1]. Here, we focus on the interactions between the targets that are considered relevant to nutrient pollution in water systems (Fig. 1). The approach to assessing the interactions between these SDGs is described in the "Methods" section. The interactions are explained in Supplementary Tables 3, 4.

Improved management in the alternative scenarios may contribute to meeting SDG 14 by reducing the potential for coastal eutrophication: we calculate lower ICEP values in the alternative scenarios compared to that in the SSP5-RCP8.5 (Fig. 8). In the AG + SE + SFC + CLI scenario, negative ICEP values are projected for the Huai, Yangtze, and Pearl Rivers, indicating their low potentials for coastal eutrophication. However, the Liao, Hai, and Yellow Rivers draining into the Bohai Gulf may still have high potential for coastal eutrophication, as indicated by positive ICEP values.

**Cobenefits of water pollution control for other SDGs.** Our future scenarios with improved water quality (aimed at meeting SDGs 6 and 14) will likely contribute to meeting SDGs 2, 11, 12, and 13, as they promote synergies and minimize tradeoffs (Supplementary Figs. 3–7) between those SDGs that are strongly related to nutrient pollution. In other words, these scenarios aim

to achieve SDGs 6 and 14 as well as SDGs 2, 11, 12, and 13 simultaneously. For instance, increased nutrient use efficiencies in agriculture, as assumed in the AG scenario, contribute to meeting SDGs 6 and 14 by reducing nutrient losses to water systems from agriculture while meeting SDG 2 "Zero Hunger" by maintaining food production. This approach avoids the tradeoffs resulting from measures (e.g., limiting fertilizer use) to improve water quality for SDG 6, which counteracts increasing agricultural productivity for SDG 2. In SE, all wastewater from rural and urban areas is assumed to be treated by sewage systems with improved nutrient removal efficiency, contributing to meeting SDG 11 "Sustainable Cities and Communities". The dietary change toward lower meat consumption and reduced food waste in the AG + SE + SFC scenario reduces the requirements for crop and animal production, which, therefore, may contribute to SDG 12 "Sustainable Consumption and Production". Finally, the climate mitigation efforts by countries underlying the AG + SE +

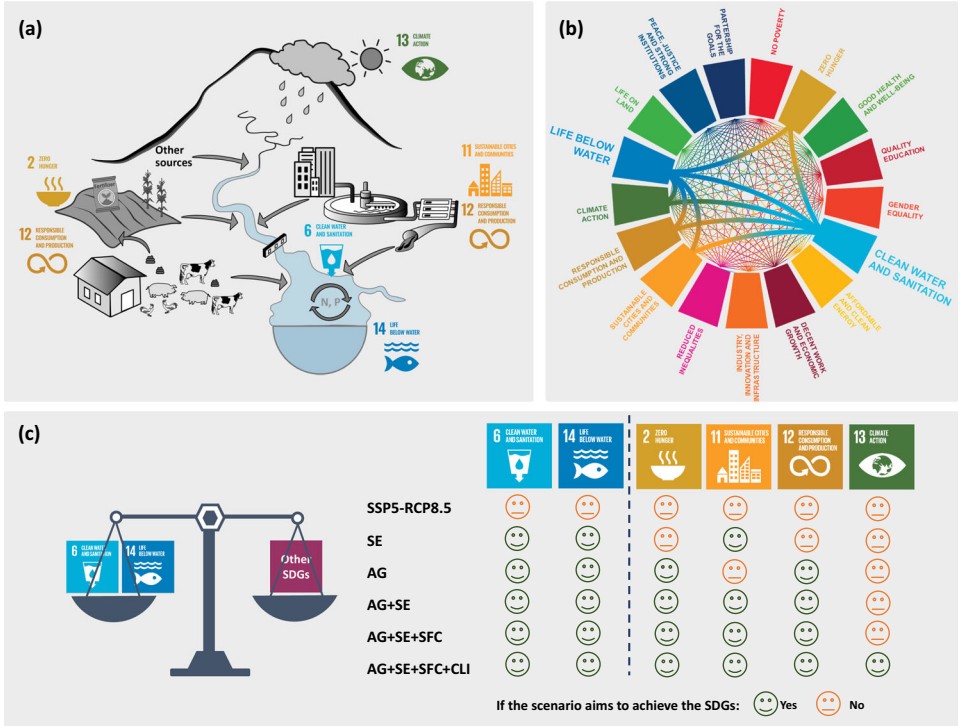

**Fig. 4 Accounting for interactions between Sustainable Development Goals (SDGs) for water pollution control in China. a** Six SDGs that are highly relevant to nutrient pollution in Chinese water systems; **b** the interactions between SDG 6 "Clean Water and Sanitation", 14 "Life Below Water" and other SDGs; **c** six scenarios that may contribute to the achievement of six SDGs that are highly relevant to nutrient pollution in Chinese water systems in 2050. The details of the interactions between SDGs are presented in Figs. 2 and 3 and Supplementary Tables 3, 4. Bold lines in **b** present the interactions between the SDGs in **a** that are highly relevant to nutrient pollution in Chinese water systems. "Yes" in **c** indicates that the scenario includes options that may contribute to meeting the SDGs. "No" in **c** indicates that the scenario does not include options that may contribute to meeting the SDGs. Six scenarios are explored: baseline SSP5-RCP8.5 (Shared Socio-economic Pathway 5-Representative Concentration Pathway 8.5), and alternative SE (improved sewage treatment), AG (improved nutrient use efficiencies in agriculture), AG + SE (a combination of AG and SE), AG + SE + SFC (sustainable food consumption in addition to AG + SE), and AG + SE + SFC + CLI (climate mitigation in addition to AG + SE + SFC) scenarios.

SFC + CLI scenario contribute to achieving SDG 13 "Climate Action".

Moreover, these scenarios may contribute to SDGs other than the six discussed thus far (SDGs 6, 4, 12, 11, 12, and 13). For example, improved fertilizer and manure management that mitigates $NH_3$ and nitrous oxide ($N_2O$) emissions may also reduce atmospheric N deposition, thus likely reducing the threats to biodiversity and contributing to SDG 15 "Life on Land". This reduced water pollution by nutrients likely also indirectly benefits livelihoods by improving environmental quality (SDG 1 "No Poverty") and illness from water contamination (SDG 3 "Good Health and Well-being").

## Discussion

Our study identifies the interactions (286 synergies and 33 tradeoffs) between the SDGs that are relevant to nutrient pollution in water systems in China. To do so, we use the seven-point-scale framework of Griggs, et al.[25]. Among the approaches for assessing the interactions between SDGs[26–28], a strength of this framework is that it allows for the quantitative assessment of the interactions between SDGs on a −3/+3 scale, providing a comprehensive understanding of how SDGs are related and helping policy makers and stakeholders set priorities for environmental management. We realize that the scoring of the interactions is inevitably, to some extent, subjective because it is in part based on interpretation. We combine scientific evidence and expert judgments to assess these interactions. When another team conducts the same process, the resulting scores may be different, depending on their interpretations of the scores and the growing scientific

evidence on these interactions. We consider this, however, a strength, as long as the assessment is transparent, as has been done in this study (see the scores and explanations in Supplementary Tables 3, 4). This approach allows other scientists and stakeholders to take our assessment of interactions and further improve it based on new scientific literature and their own expertise.

The interactions between SDGs are identified for China. The number of interactions is largely generic for other countries or regions. However, the scoring of many interactions, ranging from −3 to +3, is specific to China. These scores may be different in other countries or regions. For example, the strong tradeoffs between food production (SDG 2.3) and water quality (SDG 6.3) have gained considerable attention in the policy agendas of China, due mainly to excessive fertilizer use and direct discharges of animal manure to surface waters without proper treatment, which is unique to China and less prominent in other countries today. The interactions identified in our study can thus be used as a basis and need to be contextualized (e.g., considering their spatial, governance, technology, and timeframe dependencies) when applying them to other countries.

Our study is unique in that it accounts quantitatively for SDG interactions in future scenarios for nutrient pollution control in the rivers and coastal waters of China. For this, we implement the MARINA 2.0 model, which was developed based on the MARINA 1.0 model that has been validated for six large rivers in China[29]. The performance of the MARINA 2.0 model is assessed by comparing the model results with observations and other models (see Figure SD1 in the SI of Wang, et al.[16]). Three

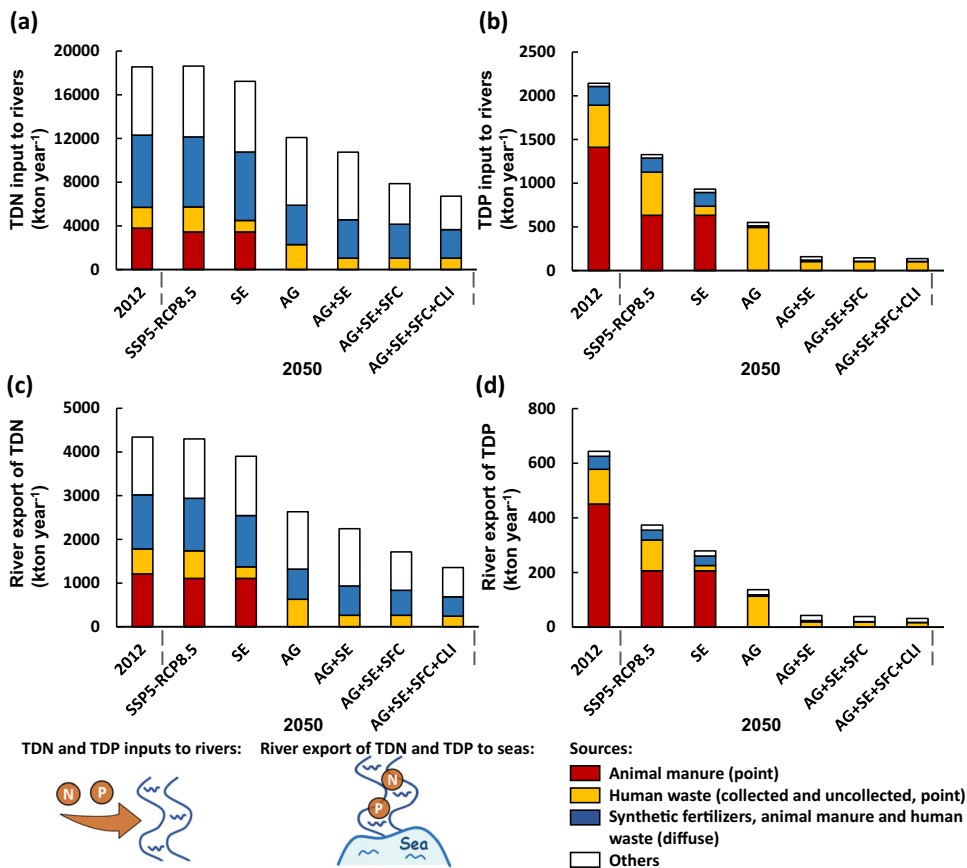

**Fig. 5 Nitrogen (N) and phosphorus (P) inputs to rivers, and river export of N and P to seas by source for six large Chinese rivers in 2012 and 2050 (kton year⁻¹). a** total dissolved N (TDN) inputs to rivers, **b** total dissolved P (TDP) inputs to rivers, **c** river export of TDN to seas, and **d** river export of TDP to seas. These rivers are the Liao, Hai, Yellow, Huai, Yangtze, and Pearl Rivers. For 2050, six scenarios are explored: baseline SSP5-RCP8.5 (Shared Socio-economic Pathway 5-Representative Concentration Pathway 8.5), and alternative SE (improved sewage treatment), AG (improved nutrient use efficiencies in agriculture), AG + SE (a combination of AG and SE), AG + SE + SFC (sustainable food consumption in addition to AG + SE), and AG + SE + SFC + CLI (climate mitigation in addition to AG + SE + SFC) scenarios. Others include biological $N_2$ fixation, atmospheric N deposition, and leaching of organic matter for TDN, as well as P weathering and leaching of organic matter for TDP.

indicators for model performance are calculated: Pearson's coefficient of determination ($R_p^2$), the Nash-Sutcliffe efficiency (NSE), and the root mean square error to the standard deviation of measured data (RSR). The model shows good performance with an $R_p^2$ of 0.85, NSE of 0.72 and RSR of 0.53 based on the performance ratings of watershed models from Moriasi et al.[30]

Nevertheless, we realize that the model results inevitably have uncertainties, which also holds for our assumptions regarding future trends. For example, we assume that pollution control options are implemented to their full technical potential in our scenarios. For sewage treatment, this assumption is based on existing technologies[22,31]. Likewise, we assume, in the AG scenario, that all manure is recycled or treated using existing technologies such as anaerobic digestion and composting[32–36]. To investigate the impacts of uncertainties in the quantitative assumptions on the effectiveness (benefits) of the scenarios in terms of pollution control, we perform a sensitivity analysis for 12 model inputs that are affected by the quantitative assumptions in our alternative scenarios (Supplementary Table 11). We change these model inputs by +10% and −10% under the AG + SE + SFC + CLI scenario, which includes all options for pollution control in our scenarios. This approach results in 24 model runs for 26 Chinese subbasins, the results of which can be found in Supplementary Tables 12–23.

We find that the benefits of our scenarios for pollution control are, in general, not largely affected by the variations in the

quantitative assumptions. The results of the sensitivity analysis show that +10% or −10% changes in 8 out of the 12 model inputs result in <5% changes in the model outputs on nutrient inputs to rivers and the river export of nutrients to seas for the sum of all subbasins and among most individual subbasins (Supplementary Tables 12–23). The model output for river and coastal water pollution (sum of 26 subbasins) is somewhat more sensitive (up to 9% changes) to changes in river discharge (Supplementary Tables 12–15). Model outputs are most sensitive to changes in three model inputs—(1) population with sewage connections, (2) removal fractions of nutrients during treatment in sewage systems, and (3) inputs of nutrients in animal manure to agricultural land—which are calculated for both the sum of all subbasins (dark orange rows indicating changes in model outputs >10% in Supplementary Tables 12–15) and individual subbasins (number of subbasins with changes in model outputs >10% in Supplementary Tables 16–23), indicating the importance of improving sewage systems and animal manure management for reducing nutrient pollution in Chinese rivers. Thus, when formulating future policies based on our scenarios, special attention needs to be paid to sewage systems and manure management regarding their economic, societal, institutional and technical feasibility in ensuring the effectiveness of the policies in pollution control.

Our scenarios and sensitivity analysis show that it is technically possible to reduce water pollution in China. We consider only

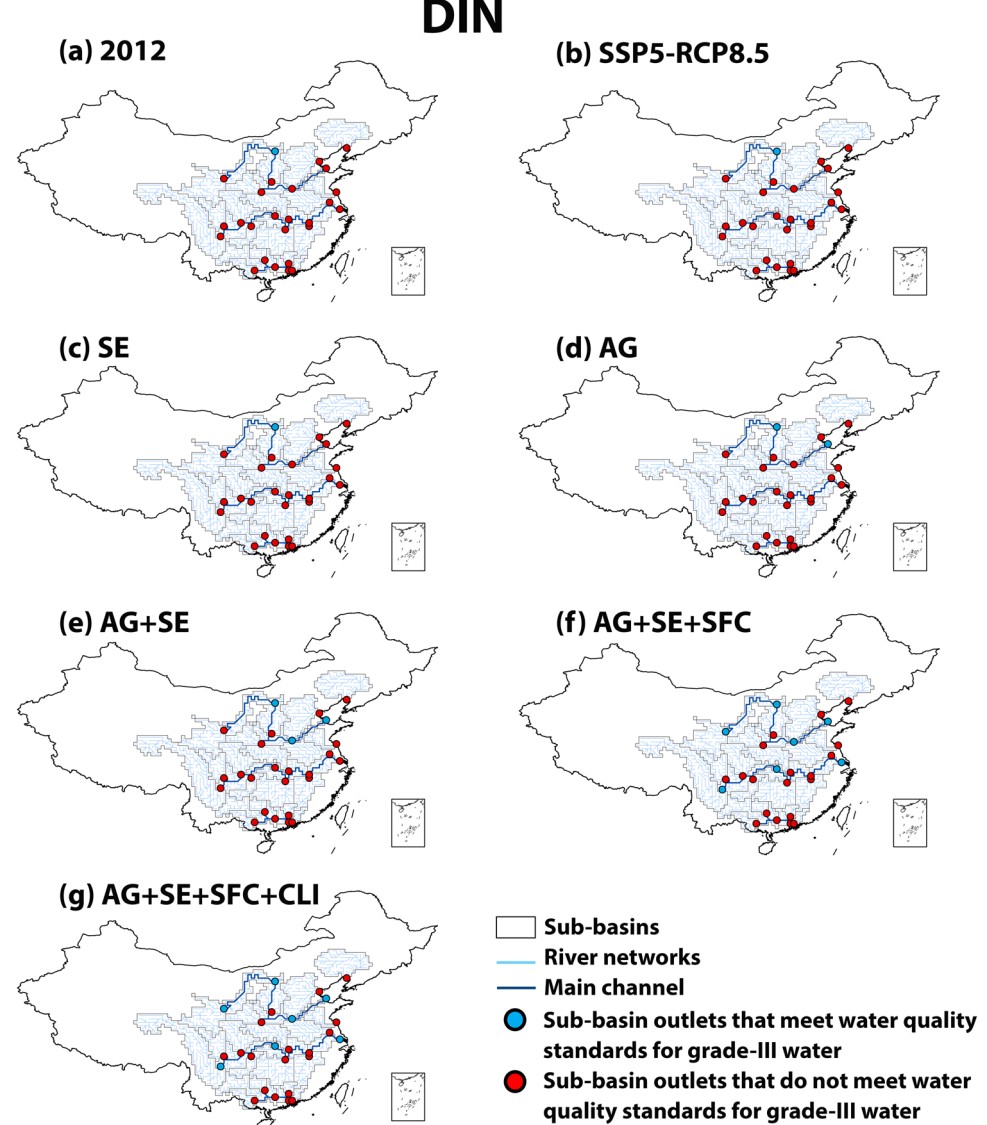

**Fig. 6 Concentrations of dissolved inorganic nitrogen (DIN) at the outlets of subbasins versus the water quality standards for "grade-III" water in China (concentrations of NH₃ < 1 mg N/L).** This comparison assesses whether SDG 6 "Clean Water and Sanitation" is met at the subbasin outlets in 2012 (**a**) and 2050 (**b**–**g**). The water quality standards for "grade-III" water are from the Chinese "Environmental Quality Standard for Surface Water"[23]. For 2050, six scenarios are explored: **b** baseline SSP5-RCP8.5 (Shared Socio-economic Pathway 5-Representative Concentration Pathway 8.5) and alternative **c** SE (improved sewage treatment), **d** AG (improved nutrient use efficiencies in agriculture), **e** AG + SE (a combination of AG and SE), **f** AG + SE + SFC (sustainable food consumption in addition to AG + SE), and **g** AG + SE + SFC + CLI (climate mitigation in addition to AG + SE + SFC) scenarios.

existing technologies and do not account for possible new developments. Since China's economy is expected to develop rapidly in the coming decades, there will likely be opportunities for the development of advanced technologies[15]. Moreover, our scenarios mainly include measures that address the sources (e.g., agriculture and sewage systems) of nutrient pollution in waters. Other measures, such as nature-based solutions or green infrastructure, can also be adopted, as they may benefit both water pollution control and other SDGs. For example, nature-based solutions such as harvesting aquatic plants from polluted rivers and lakes can help reduce eutrophication in waters (SDG 6) and recover nutrients for the production of fertilizers and animal feeds (SDG 2)[37,38]. Therefore, we consider our scenario challenging to achieve but not impossible to realize by 2050 when also considering the current ambitious aim of meeting the SDGs by 2030.

Our analyses show the importance of appropriate indicators for SDGs 6 and 14. Following the current national plan regarding the implementation of the SDGs[18], we use the Chinese "Environmental Quality Standard for Surface Water" for "grade-III" water as a standard for good ambient water to assess SDG 6. According to this standard, the concentration of NH₃ in rivers may not exceed 1.0 mg-N/L, and that of TP may not exceed 0.2 mg-P/L. This standard is from "China's National Plan on Implementation of the 2030 Agenda for Sustainable Development", which was developed to implement the UN's SDG targets in the Chinese context. We thus consider this standard a useful indicator that represents SDG target 6.3 well for China.

This standard, however, does not provide information on the concentrations of other N and P forms (e.g., nitrite ($NO_2^-$), nitrate ($NO_3^-$), or total N), which may lead to a biased assessment of water quality in practice. For example, water in some river sections may have low concentrations of NH₃ but high concentrations of other forms of N, which can lead to eutrophication. Water quality in these river sections might then be

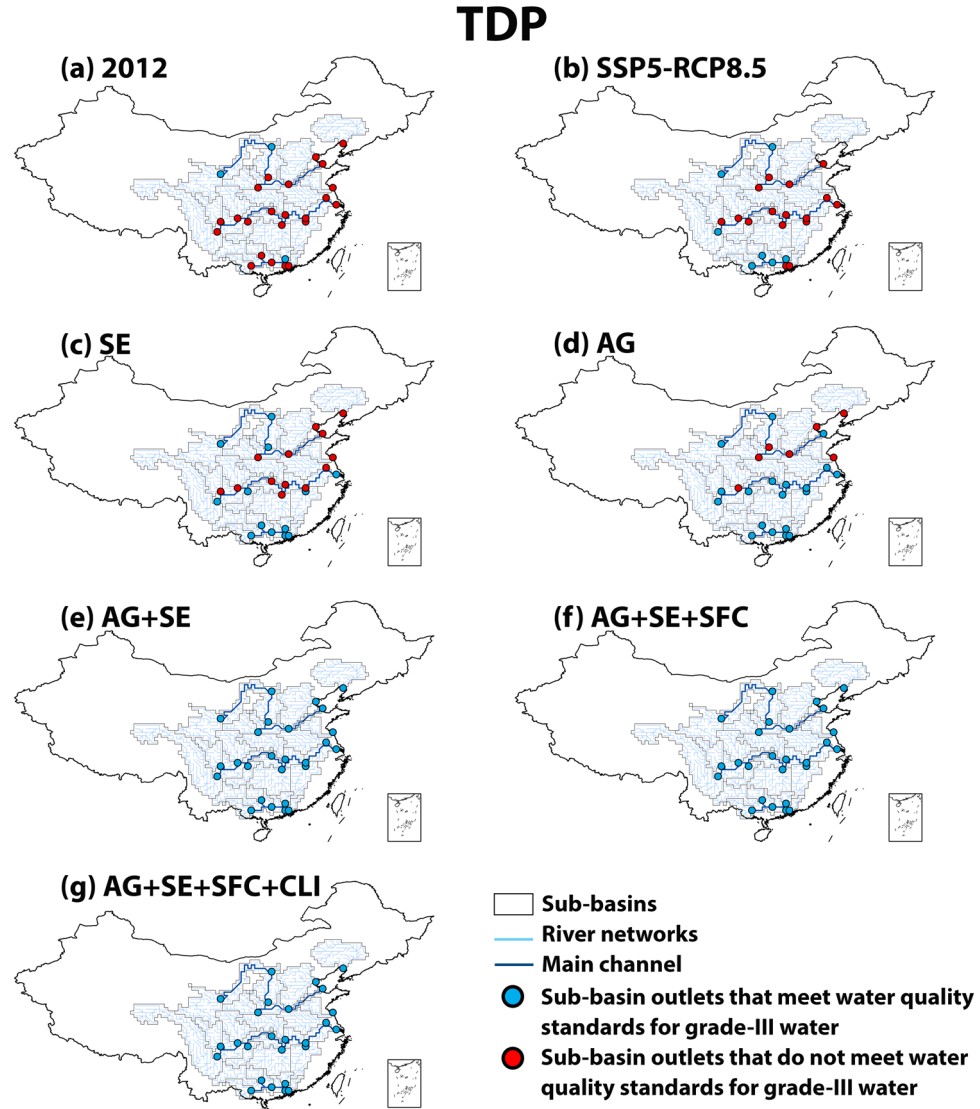

**Fig. 7 Concentrations of total dissolved phosphorus (TDP) at the outlets of subbasins versus the water quality standards for "grade-III" water in China (concentrations of TP < 0.2 mg P/L).** This comparison assesses whether SDG 6 "Clean Water and Sanitation" is met at the subbasin outlets in 2012 (**a**) and 2050 (**b**–**g**). The water quality standards for "grade-III" water are from the Chinese "Environmental Quality Standard for Surface Water"[23]. For 2050, six scenarios are explored: **b** baseline SSP5-RCP8.5 (Shared Socio-economic Pathway 5-Representative Concentration Pathway 8.5) and alternative **c** SE (improved sewage treatment), **d** AG (improved nutrient use efficiencies in agriculture), **e** AG + SE (a combination of AG and SE), **f** AG + SE + SFC (sustainable food consumption in addition to AG + SE), and **g** AG + SE + SFC + CLI (climate mitigation in addition to AG + SE + SFC) scenarios.

qualified as good if based only on $NH_3$ concentrations. Therefore, appropriate indicators for river pollution that consider other nutrient forms are needed. Such indicators can be developed by improving the existing water quality standard for water bodies based on their purpose of use. Examples of such standards from other countries (e.g., the United States and the European Union) are summarized in Supplementary Table 9. Additionally, many water scarcity indicators[39–41], such as those from Liu et al.[40] and van Vliet et al.[41,42] based on the supply and demand of water of suitable quality, can also be useful for assessing the progress toward achieving SDG 6 in China.

Like many countries, China does not have a specific indicator to monitor the changes in coastal eutrophication for SDG 14. Our study uses ICEP as an index for assessing the potential of coastal eutrophication. ICEP has been widely accepted and used in global[43] and regional studies to assess coastal eutrophication[16,29,44,45]. Moreover, it has been recently accepted by the UN Environment Programme as an index for assessing

coastal eutrophication for SDG 14.1[46,47]. We thus consider ICEP a representative indicator for assessing coastal eutrophication for SDG 14.1 in China. We realize that ICEP has its limitations. Most importantly, it is calculated on an annual basis and does not consider seasonality in coastal eutrophication. Thus, ICEP needs to be interpreted with care. Positive ICEP values indicate a high potential for harmful algae blooms, but negative ICEP values do not guarantee the absence of harmful algae blooms. Nevertheless, we consider ICEP a useful first step toward assessments of SDG 14.

Our results contribute to policies on sustainable development. "China's National Plan on Implementation of the 2030 Agenda for Sustainable Development"[48] lays out specific plans for the implementation of the 17 SDGs and their 169 targets. Accounting for our identified tradeoffs and synergies between SDGs can certainly contribute to this national plan to achieve sustainability for all. Policies that ignore tradeoffs and synergies may meet some SDGs at the expense of others. One example is the recent Chinese

| River | Sea | ICEP (kg C-eq. km⁻² day⁻¹) | | | | | | |
|---|---|---|---|---|---|---|---|---|
| | | | 2050 | | | | | |
| | | 2012 | SSP5-RCP8.5 | SE | AG | AG+SE | AG+SE+SFC | AG+SE+SFC+CLI |
| Liao | | 14 | 12 | 11 | 4 | 3 | 2 | 2 |
| Hai | | 21 | 21 | 17 | 10 | 7 | 6 | 5 |
| Yellow | Bohai Gulf | 2 | 2 | 1.4 | 0.9 | 0.6 | 0.6 | 0.4 |
| Huai | Yellow Sea | 51 | 45 | 33 | 12 | -0.1 | -0.2 | -0.5 |
| Yangtze | East China Sea | 16 | 7 | 4 | -0.5 | -3 | -3 | -3 |
| Pearl | South China Sea | 18 | 10 | 5 | 2 | -3 | -4 | -4 |

ICEP > 1 (red) / -1 < ICEP < +1 (yellow) / ICEP < -1 (green)

**Fig. 8 Indicator for Coastal Eutrophication Potential (ICEP) for the six large rivers in China in 2012 and 2050 (kg C-eq. km$^{-2}$ day$^{-1}$).** ICEP is used to assess whether SDG 14 "Life Below Water" is met at the river mouths. A positive ICEP value indicates that river export of nutrients is in excess over silica (Si), and thus indicates high potential for coastal eutrophication. A negative ICEP value indicates low potential for coastal eutrophication. Following the approach of ref. [43], we calculate ICEPs based on our modeled river export of dissolved nitrogen and phosphorus, and river export of Si from the Global NEWS-2 model[68]. For 2050, six scenarios are explored: baseline SSP5-RCP8.5 (Shared Socio-economic Pathway 5-Representative Concentration Pathway 8.5) and alternative SE (improved sewage treatment), AG (improved nutrient use efficiencies in agriculture), AG + SE (a combination of AG and SE), AG + SE + SFC (sustainable food consumption in addition to AG + SE), and AG + SE + SFC + CLI (climate mitigation in addition to AG + SE + SFC) scenarios.

policy on banning livestock production in so-called nonlivestock production regions (NLPRs)[49,50]. This policy aims to control water pollution in vulnerable water bodies by moving pig production from NLPRs to southwest and northeast provinces. This policy is estimated to reduce nutrient losses to waters by up to 27% for N and 48% for P in Southeast China, contributing to meeting SDG 6 in NLPRs in China[51]. However, this policy may increase N losses to the air in the southeast and northeast, which may lead to biodiversity loss, thus having adverse impacts on SDG 15 "Biodiversity on Land"[51]. Moreover, this policy may also increase N loss to groundwater in the southeast and northeast because of the ignorance of the spatial tradeoffs between regions (South China versus Southeast and Northeast China) toward achieving SDG 6. Thus, it is important to account for the interactions between SDGs across sectors and regions in future policies to increase positive synergy effects and avoid the adverse impacts of SDGs 6 and 14 on the other SDGs. Our most effective scenario, AG + SE + SFC + CLI, is an example of how future policies can contribute simultaneously to the achievement of SDGs 6 and 14 and to that of the other SDGs.

Several important aspects need to be considered to effectively account for our identified interactions in future policies. First, in our study, the cobenefits of water pollution control are, however, assessed based on the assumptions in the scenarios and not by the model (see "Methods"). This is because integrated modeling frameworks that combine models from different disciplines (e.g., food, water, hydrological, and economic disciplines) are scarce. Such modeling frameworks are strongly needed, as they can contribute to (1) assessing multiple SDGs under consistent climate and socioeconomic conditions, (2) quantifying SDG interactions and (3) supporting decision makers by modeling the effectiveness of accounting for SDG interactions in their policies. Thus, to better account for interactions in future policies, further research is needed to develop and apply system (e.g., integrated modeling) approaches to analyze SDG interactions. Furthermore, we take the identified interactions as a "checklist" to check whether our scenarios for water pollution control promote synergies or mitigate tradeoffs between SDGs. There are other approaches to account for SDG interactions in future policies. For example, decision makers can start from the list of interactions and select and develop policies for those they would like to address. To do this effectively, important questions, such as "Should priorities be placed on synergies of higher scores?" and

"How can promoting synergies and mitigating tradeoffs be balanced if they cannot be achieved simultaneously?" need to be answered given the regional context.

Our work shows the importance of accounting for the interactions between SDGs in water pollution control for China. China has many climate regimes and land use types. Thus, many of these interactions between water-pollution-related SDGs (e.g., SDG 2 "Zero Hunger" and SDG 6 "Clean Water and Sanitation") may exist in many regions of the world. The identified interactions for China provide a good basis for analyzing them in other countries while adjusting for local situations. The effective scenarios for water pollution control from this work may also guide other countries in dealing with water pollution while also achieving sustainability in other domains.

## Methods

**Identifying SDGs related to nutrient pollution in Chinese water systems**. The SDGs (and their targets) that are relevant to nutrient pollution in Chinese water systems are identified based on the existing literature (Supplementary Table 1) and expert judgments. The targets of the 17 SDGs are officially listed as one-sentence statements that guide SDG implementation. Based on these one-sentence statements, we identify keywords for each SDG target (Supplementary Table 5). In this way, the potential link between a target and nutrient pollution is investigated by performing a keyword search in the existing literature. The keywords for each target are compared to different keywords related to nutrient pollution, such as "nutrient pollution", "nutrient management", or "water quality", to span the array of academic literature that potentially exists on the subject. Additional keywords such as "China" or "Chinese waters" are added to the query, so the literature review is made specific to the national or regional context. In the case where no specific information is found, information is then extrapolated from global studies (Supplementary Table 1). We define three levels of the relevance of targets to nutrient pollution: "high", "moderate" and "low". The level referred to as "high" encompasses the targets that address the direct sources of nutrient losses to the Chinese water systems. The targets identified as being of "moderate" relevance comprise factors that address the impacts of nutrient pollution on aquatic ecosystems and human health or influence the resilience of ecosystems to nutrient pollution. The targets of "low" relevance cover technological, social, administrative or economic interventions indirectly related to nutrient pollution in water systems. The targets identified as "high", "moderate" and "low" are listed in Fig. 1 and Supplementary Table 1.

**Assessing synergies and tradeoffs between SDGs 6 and 14 and other SDGs**. The target-level interactions between SDGs relevant to nutrient pollution in inland freshwater and coastal waters in China are assessed. More specifically, an innovative aspect of this research lies in its evaluation of the potential positive (synergetic) and negative (tradeoff) interactions existing between SDG targets in the context of nutrient pollution in Chinese water systems. These interactions are assessed based on the existing literature (Supplementary Table 1) and our expert judgments on water pollution in China using the seven-point-scale framework of

Griggs et al.[25], who classified the interactions at 7 levels: (−3) canceling, (−2) counteracting, (−1) constraining, (0) consistent, (+1) enabling, (+2) reinforcing, and (+3) indivisible. The negative levels refer to tradeoffs, while the positive levels refer to synergies. Zero refers to a neutral relationship between targets. The definitions of the 7 levels of interactions are summarized in Supplementary Table 2.

The identified interactions have a direction, either unidirectional (one direction) or bidirectional (two directions). A unidirectional interaction means that target A affects target B, but target B does not affect target A, whereas a bidirectional interaction means that target A affects target B, and target B also affects target A. An example of a unidirectional interaction is the tradeoff between targets 2.3 and 6.4: water use in intensive agriculture to meet target 2.3, aiming to double agricultural productivity, may counteract reducing water scarcity to meet target 6.4. An example of bidirectional interaction is the synergy between targets 11.6 and 6.3: reducing water pollution in cities by improving wastewater management to meet target 11.6 is indivisible from improving water quality by halving the proportion of untreated wastewater to meet target 6.3, and vice versa. The identified interactions and their directions are illustrated in Figs. 2 and 3 and explained in Supplementary Tables 3, 4. We realize that such an assessment of the interactions can differ among experts and therefore require continuous iterations and improvements. The interactions that we identified, however, provide a primary and good basis for such continuous effort that contributes to understanding how SDGs are interrelated in the context of water pollution in China.

**Scenarios.** We explore future (1 baseline + 5 alternative) scenarios to achieve the SDGs for improved river and coastal water quality in China using the MARINA 2.0 model. Our five alternative scenarios are developed to reduce water pollution while benefitting agriculture, sewage, food consumption, and climate mitigation by accounting for the interactions between the SDGs. We account for synergies and tradeoffs in developing these scenarios through the following steps. First, we make an inventory of the measures that are effective in reducing nutrient pollution in Chinese water systems based on existing scenario analyses[52–55]. Next, based on the identified SDG interactions, we identify the measures that contribute to achieving SDGs 6 and 14 as well as SDGs 2, 11, 12, and 13 simultaneously. In other words, we try to include in our scenarios only the measures that promote synergies and avoid tradeoffs between SDGs 6 and 14 and SDGs 2, 11, 12, and 13. For example, agricultural practices and technologies to improve nutrient use efficiencies are adopted in the alternative scenarios, which reduces nutrient losses from agriculture for SDGs 6 and 14 while maintaining food production for SDG 2 (synergies between SDGs). Measures to control water pollution, such as reducing fertilizer use, which may result in yield losses, are thus not considered, as they can lead to challenges in achieving SDG 2 (tradeoffs between SDGs). In other words, the five alternative scenarios are developed based on measures of action promoting the synergies and mitigating the tradeoffs between key SDGs (i.e., water, agriculture, sewage, food consumption, and climate mitigation) (Supplementary Table 6). The interactions (synergies and tradeoffs) addressed by each specific assumption in the alternative scenarios are presented in Supplementary Table 7 and Supplementary Figs. 3–7.

The baseline SSP5-RCP8.5 scenario assumes relatively low population growth, fast economic growth, high fossil fuel consumption, and high international trade, increasing productivity in agriculture and environmental policies for local issues[16,56,57]. As a result, in 2050, sewage systems will be slightly improved compared to those today. Not all wastewater will be connected to sewage systems, especially in rural areas, where only 10% of wastewater will be collected (Supplementary Table 6). Nutrient removal during treatment will remain low or moderate at ~12–47% for N and 44–75% for P in rural and urban areas (Supplementary Table 6). Crops will be produced with fewer resources (e.g., nutrients, land, and water) because of increased productivity. Animal production will be intensive and industrialized to meet the increasing preference for meat-rich diets. Improved manure management is implemented to reduce emissions of $NH_3$ and $N_2O$ during manure storage and housing. A total of 15–41% of crop residues and 70% of animal manure will be recycled in agriculture (Supplementary Table 6). The remainder will be lost to the environment. The import of food for consumption will be 17% higher in 2050 than in 2012 (Supplementary Table 6). The greenhouse gas (GHG) emissions of China, as well as those of other countries, will be high due to high fossil fuel consumption.

The SE (improved sewage treatment) scenario builds on the SSP5-RCP8.5 and assumes further improved sewage systems by 2050 based on the targets of SDG 11 "Sustainable Cities and Communities". According to current Chinese policies, wastewater connected to sewage systems will reach 70–95% in urban areas and 60% in one-third of China's counties, including rural and urban areas, by 2050 (Supplementary Table 6). We, therefore, assume in this scenario that by 2050, all wastewater will be connected to centralized (in urban areas) or decentralized (in rural areas) sewage systems, following Strokal et al.[52] Nutrient removal during treatment is assumed to reach 80% for N and 90% for P by adopting the best treatment technologies[22,52] (Supplementary Table 6). These scenario assumptions promote 9 synergies and mitigate 3 tradeoffs between SDGs for clean water (SDGs 6 and 14) and SDG 11 (Supplementary Fig. 3).

The AG (improved nutrient use efficiencies in agriculture) scenario builds on the SSP5-RCP8.5 and assumes further improved nutrient use efficiencies in agriculture by 2050 based on the targets of SDG 2 "Zero Hunger". In this scenario,

crops will be fertilized according to their needs for nutrients based on a balanced fertilization approach[53,54]. As a result, the use of synthetic fertilizers will be largely reduced compared to the baseline, without yield loss. Recycling up to 80% of straw residues on cropland will largely reduce air pollution due to straw burning (Supplementary Table 6). Animal production will be more efficient by using improved animal feeding and genetically modified animals that use nutrients more efficiently[58]. In the AG scenario, N and P excretions are thus 12% lower than in the baseline SSP5-RCP8.5 (Supplementary Table 6). Improved manure management is incorporated to reduce $NH_3$ and $N_2O$ emissions during manure storage[59–61]. In the AG scenario, the direct discharge of manure will be restricted by policies; thus, all manure is assumed to be treated and recycled on cropland. These scenario assumptions promote 8 synergies and mitigate 10 tradeoffs between SDGs for clean water (SDG 6 and 14) and SDG 2 (Supplementary Fig. 4).

The AG + SE scenario combines the storylines of the SE and AG scenarios that are developed based on SDGs 2 and 11. The AG + SE scenario assumes improved sewage systems and nutrient use efficiencies in agriculture. This scenario will promote 17 synergies and mitigate 13 tradeoffs between SDGs for clean water (SDG 6 and 14) and SDGs 2 and 11 (Supplementary Fig. 5).

The AG + SE + SFC (sustainable food consumption in addition to AG + SE) scenario builds on AG + SE scenario and assumes additionally healthier diets and less food waste by 2050 based on the targets of SDG 12 "Responsible Consumption and Production". In this scenario, society will follow Chinese dietary guidelines (CDGs)[62], which recommend consuming less meat and more milk, eggs, vegetables, and fruits. Food waste will be reduced by 20% through responsible consumption, improved food processing, and storage facilities[55]. The reduction in meat consumption and food waste will result in a 20% reduction in the requirements for crop and animal production. China may remain a large importer of soybean due to limited land resources and increasing food demand[63]. For soybeans, we assume that approximately 80% of the soybean consumption in 2050 will be imported from abroad, following the assumption in Ma et al.[55] In addition to the above assumptions, this scenario assumes the further improved management of animal manure. In the AG + SE scenario, many river basins do not have enough arable land to recycle all the manure produced in the basin. Therefore, the AG + SE + SFC scenario assumes that the excessive manure will be either treated (as effectively as wastewater) or exported to other regions in China to be recycled. Finally, atmospheric N deposition is assumed to be reduced by 50% relative to that in the SSP5-RCP8.5 by reducing $NH_3$ and nitrogen oxide ($NO_x$) emissions in the agricultural and nonagricultural sectors (e.g., controlling $NH_3$ and $NO_x$ emissions from industries). These scenario assumptions promote 42 synergies and mitigate 13 tradeoffs between SDGs for clean water (SDGs 6 and 14) and SDGs 2, 11, and 12 (Supplementary Fig. 6).

The AG + SE + SFC + CLI (climate mitigation in addition to AG + SE + SFC) scenario builds on the AG + SE + SFC scenario and additionally assumes a global effort in climate mitigation by 2050 based on the targets of SDG 13 "Climate Action". In an earlier study using the MARINA 2.0 model[16], the baseline SSP5-RCP8.5 scenario assumes high GHG concentrations under higher fossil fuel consumption, which will lead to considerable climate change and thus affect hydrology (e.g., river discharge). The AG + SE + SFC + CLI scenario assumes that GHG emissions will be reduced to the level of the RCP2.6 scenario by 2050, which implies efforts by countries worldwide to reduce GHG emissions to achieve Paris Agreement temperature targets[64]. The lower GHG emissions in the future may result in fewer increases in precipitation and river discharge than in the baseline, thus lessening the decrease in the in-river retention of nutrients. The river export of nutrients may thus be reduced by climate mitigation in this scenario compared to the baseline. These scenario assumptions promote 56 synergies and mitigate 13 tradeoffs between SDGs for clean water (SDGs 6 and 14) and SDGs 2, 11, 12, and 13 (Supplementary Fig. 7).

**MARINA 2.0 model.** We use the MARINA 2.0 model[16] to explore future nutrient pollution in the rivers and coastal waters of China. This model is developed to quantify the river export of TDN and TDP in four forms by rivers at the subbasin scale from different sources[16]. The four nutrient forms are dissolved inorganic N (DIN), dissolved organic N (DON), dissolved inorganic P (DIP), and dissolved organic P (DOP). TDN is the sum of DIN and DON, and TDP is the sum of DIP and DOP.

The MARINA 2.0 model quantifies the river export of TDN and TDP as a function of N and P inputs to surface waters (rivers) from diffuse and point sources and retentions of N and P in rivers based on Eq. 1, respectively[16,29]:

$$M_{F,y,j} = (RSdif_{F,y,j} + RSpnt_{F,y,j}) \cdot FE_{riv.F.outlet.j} \cdot FE_{riv.F.mouth.j} \qquad (1)$$

where $M_{F,y,j}$ is the river export of N and P in form $F$ (DIN, DON, DIP, DOP) by source $y$ from subbasin $j$ (kg year$^{-1}$). $RSdif_{F,y,j}$ is the N and P inputs in form $F$ to rivers (surface waters) from diffuse sources $y$ in subbasin $j$ (kg year$^{-1}$). $RSpnt_{F,y,j}$ is the N and P inputs in form $F$ to rivers from point sources $y$ in subbasin $j$ (kg year$^{-1}$). $FE_{riv.F.outlet.j}$ is the fraction of N and P in form $F$ exported to the outlet of subbasin $j$ (0–1). $FE_{riv.F.mouth.j}$ is the fraction of N and P in form $F$ exported from the outlet of subbasin $j$ to the river mouth (0–1). The detailed equations to quantify $RSdif_{F,y,j}$, $RSpnt_{F,y,j}$, $FE_{riv.F.outlet.j}$ and $FE_{riv.F.mouth.j}$ are available in the SI of Wang et al.[16].

We model nutrient pollution in the rivers and coastal waters of six large rivers in China (Supplementary Fig. 1). These rivers include the Liao, Hai, and Yellow Rivers draining into the Bohai Gulf; the Huai River draining into the Yellow Sea; the Yangtze River draining into the East China Sea; and the Pearl River draining into the South China Sea. We select these rivers because they contribute largely to nutrient pollution in the coastal waters of China. According to Wang et al.[16], these six rivers contributed ~90% to the river export of TDN and TDP to the Chinese seas in 2012. The drainage basins of the Yellow, Yangtze, and Pearl Rivers are divided into upstream, middle-stream and downstream subbasins, respectively, following Wang et al.[16] The names of the subbasins are available in Supplementary Fig. 2.

**Indicators for SDGs 6 and 14.** Two indicators are calculated from the MARINA 2.0 model results to assess whether SDGs 6 and 14 are met. We use water quality standards for N and P concentrations as the indicator for SDG 6 and the Indicator for Coastal Eutrophication Potential (ICEP) for SDG14. Below, we describe how these indicators are chosen based on the UN-defined indicators and how they are calculated.

The goal of SDG 6 is to "ensure the availability and sustainable management of water and sanitation for all"[65]. One important indicator for assessing SDG 6 is the "6.3.2 proportion of bodies of water with good ambient water quality", according to the global indicator list from the UN[66]. In this study, we take an indicator for "good ambient water quality" from the Chinese "Environmental Quality Standard for Surface Water"[23]. This standard was adopted by "China's National Plan on Implementation of the 2030 Agenda for Sustainable Development" to achieve SDG 6[18]. China developed this plan to translate each target of the SDGs into "action plans", considering opportunities and challenges that it faces in implementing the 2030 Agenda. According to the Chinese "Environmental Quality Standard for Surface Water", "third grade" (grade III) refers to good ambient water quality[23]. For "grade-III" water in rivers, the concentration of $NH_3$ may not exceed 1.0 mg-N/L, and that of total P (TP) may not exceed 0.2 mg-P/L. The MARINA 2.0 model quantifies DIN (including $NH_3$, $NO_3^-$, and $NO_2$) and TDP but not $NH_3$ and TP. Therefore, we calculate N and P concentrations at the outlets of subbasins using modeled DIN and TDP loads and river discharges at the outlets. We compare the calculated concentrations of DIN and TDP with the water quality standards for "grade-III" water and discuss whether our scenarios contribute to the achievement of SDG 6.

The goal of SDG 14 is to "conserve and sustainably use the oceans, seas and marine resources for sustainable development"[65]. The UN's global indicator list suggests "14.1.1 Index of Coastal Eutrophication" as an indicator for this SDG[66]. Therefore, we take ICEP as an indicator for assessing the potential of coastal eutrophication for SDG 14, as it indicates the potential for the new production of harmful algae in coastal waters. This indicator is calculated by comparing the N, P, and silica (Si) loads and the Redfield molar ratios (C:N:P:Si ratios: 106:16:1:20) (see Garnier et al.[43] for the detailed approach to quantifying the ICEP). Positive ICEP values indicate relatively high potentials for harmful algal blooms when rivers deliver excess N or P over Si to the sea. Negative ICEP values indicate relatively low potentials for harmful algal blooms. We calculate the ICEP values for the six Chinese rivers using the modeled river export of TDN and TDP from the MARINA 2.0 model. Based on the results, we discuss whether our scenarios contribute to the achievement of SDG 14.

## Data availability

All data on the SDG interactions generated and analyzed in this study are included in this published article and its Supplementary information files. The datasets of the scenario analysis and sensitivity analysis generated and analyzed in this study are publicly available in the Data Archiving and Networked Services (DANS Easy) repository https://doi.org/10.17026/dans-x5w-m3q3[67].

## Code availability

All equations for the MARINA 2.0 model are provided in the Supplementary information files of the following open access publication: Wang, M., Kroeze, C., Strokal, M., van Vliet, M.T.H. & Ma, L. Global change can make coastal eutrophication control in China more difficult. *Earth's Future* 8, e2019EF001280, DOI: 10.1029/2019ef001280 (2020).

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

## Acknowledgements

We acknowledge the support of the KNAW-MOST project "Sustainable Resource Management for Adequate and Safe Food Provision (SURE+)" (PSA-SA-E-01, supporting M.W., C.K., L.M., A.J., and M.S.) and The National Key Research and Development Program of China (2016YFE0103100, supporting M.W., C.K., L.M., A.J., and M.S.). We also acknowledge the Dutch Talent Program Veni-NWO projects (VI.Veni.194.002, supporting A.J., and 0.16.Veni.198.001, supporting M.S.) and the National Natural Science Foundation of China (U20A2047, supporting L.M. and M.S.).

## Author contributions

M.W., C.K., and L.M. designed the research. M.W., A.J., and J.B. performed the assessment of the SDG interactions, which were reviewed and commented on by C.K., L.M., and M.S. M.W. performed the scenario analysis. M.W. drafted the first version of the manuscript. M.W., A.J., J.B., M.S., L.M., and C.K. contributed to the interpretation of the results, critical revision of the manuscript, and approval of the final version of the manuscript.

## Competing interests

The authors declare no competing interests.
