## [Peer Review File · Nature Communications]

Reviewer comments, first round review -

Reviewer #1 (Remarks to the Author):

This paper assesses the interactions between SDGs in terms of pollution control. It is critical and novel to investigate the use of SDG interactions to develop interventions and policies. This paper is well-written and technically sound. Some major comments are provided below.

The work has two potential contributions: 1) identified the SDGs that are relevant to nutrient pollution and assessed their interactions; and 2) developed five control scenarios and assessed their benefits in meeting the requirements. However the methodology in linking these two parts is not described. The control scenarios are reasonable and how they address the SDG interactions is also described, however, it is not clear why these scenarios are developed and if they are good control measures. It would be useful to develop a method for control measure development based on the SDG interactions. At least, clarifications should be provided on how the synergies and trade-offs between SDGs were used for developing these scenarios. For example, should priorities be placed on synergies of 'indivisible' instead of 'enabling'? how to balance promoting synergies and mitigating trade-offs if they cannot be achieved simultaneously?

With reference to the above comment on control measures, many other measures such as nature-based solutions or green infrastructure are generally thought to have multiple benefits. I wonder, how can these green measures be incorporated in scenarios considering the identified SDG synergies and trade-offs.

The interactions between SDGs in water pollution control are identified for China. Why should these interactions be limited to China? It seems in Tables A3 and A4 that general knowledge is applied to assess the interactions. It is not clear what specific data and knowledge about China are applied to the assessment.

The use of indicators is problematic. There are clearly defined indicators for each of the targets in SDGs 6 and 14 by the UN. Why are they not used and why does China need to define its own indicators? Most importantly, the two indicators used in this paper cannot represent those SDG targets well.

How can the assumptions in the scenarios affect the analysis results? In addition to the uncertainties in the MARINA model, there are many assumptions and numbers used in each scenario. I appreciate references are provided in many places, but many numbers are very subjective and could vary significantly under different conditions. Also some numbers, such as the nutrient removal rates in SE, will also vary across different regions and river basins. Some basic sensitivity analysis is needed to understand how the benefits of pollution control

are affected.

Lines 270-277, the example seems inappropriate as it shows the spatial trade-offs between the same goal (SDG6), while the previous statement is about the trade-offs between different SDGs.

It seems a section with heading 'Nutrient pollution in rivers and coastal waters' in the SI is not referenced and repeats the numbering of figures and tables.

Reviewer #2 (Remarks to the Author):

Remarks to the Author:

This paper addresses the important question of the interactions between the Sustainable Development Goals (SDGs), and its implications on water pollution control in China. I appreciate the great efforts by authors to analyze the >300 complex interactions between the SDG targets and develop simple scenarios as the first step to account for these interactions in policies for reducing future water pollution.

The result of this paper is original and is of crucial importance for controlling water pollution as a serious issue in China as well as the overall goal of sustainable development, which are prioritized in many Chinese policies these days. Many policies introduce guidelines/interventions for sustainable water, food and other sector's developments but temporally or spatially overlook the trade-offs between the sectors, which may cause failures of the policies. Understanding the interactions is the key to unlocking their full potential and ensuring that progress made in some areas is not made at the expense of progress in others. I've seen publications highlighting the importance of accounting for the SDG interactions in policies, or publications proposing approaches to analyze the interactions. A study like this by the authors, where they investigate the interactions at the detailed SDG target level in a specific perspective of nutrient pollution in the Chinese context, does not exist as far as I know.

This paper develops unique approaches based on an existing method to (i) identify the SDGs goals and targets that are relevant to nutrient pollution in waters of China, (ii) identify the synergies and trade-offs between the identified SDGs goals and targets, (iii) analyze the importance of considering the synergies and trade-offs in formulating environmental policies. I found the approaches used here is simple, elegant and transparent. A compliment to the detailed explanations for the interactions in the supplementary information (SI), which provide a transparent understanding of the interaction assessment and serve as a basis for replicating this study in another context.

Overall, I felt this paper contributes to our knowledge at a time when there is rising attention

by both scientists and policy-makers in achieving sustainable development given the complex interactions between and within nature, society and global change. The great efforts and novel results deserve to be published after some revisions given the suggestions below.

General comments:

The authors analyzed the interactions between SDGs for water pollution in China. The methods and results on SDG interactions are relatively clear. A question to the discussion of these results: are these interactions beneficial for other countries which also have problems with nutrient pollution? To what extent are these results specific for China? Or in other words, if others take these interactions for another country/region, what do they first need to check before applying them?

Another question to interaction assessment. What are the uncertainties related to this assessment? The seven-point approach involved subjectivity in any case. How does this influence the results/conclusions? Suggest adding some discussion on this.

The authors analyzed the co-benefits of water pollution control that benefits from synergies and avoids trade-offs based on scenario analysis. They indicate the interactions that can be addressed by their scenarios in the SI. To me, it is not very clear which assumptions/options in the scenarios in Table C1 address which the interactions in Figures C1-C5. Suggest making this clear.

The co-benefits of water pollution control by promoting synergies and avoiding trade-offs on other SDGs is not assessed by the model but based on the assumptions in the scenario as I understood. This is fine as the first try-out to account for the interactions. I know such assessment is complex and will need to be integrated models from different disciplines, which is challenging and is, anyway, not the main goal of this paper. However, a more elaborated discussion on how next steps/effort, especially using integrated modeling (e.g., combining food, water, hydrological, economic and etc. models), can be taken to assess the effectiveness of accounting for interactions in future policies is needed.

In the discussion, the authors pointed out an important issue: there is a general lack of quantitative indicators to monitor the progress of SDG6. This is a very good point. The authors discussed the possibilities to develop indicators based on the water quality standards in other countries, which may be challenging in my view as the standards differ largely among countries. There are existing indicators to assess water scarcity based on both quality and quantity (<https://www.nature.com/articles/ngeo3047>). Why not include them?

Specific comments:

Line 60. Please refer to the SI or Method where this seven-point scale approach is described. The report is rather long for authors to find their description of the approach.

Line 87. “nutrient pollution in waters” -> “water pollution by nutrients”

Line 118-120. What negative side-effects? Could you provide some examples to make it more clear?

Line 133. Also consider the ongoing pandemic?

Line 159. Reference needed for China’s water quality standard.

Line 228. Very good that the authors show the quantitative ratings for the model assessment

Line 350-354. See my comments above, which assumptions address which interactions should be made more explicit, at least in the SI.

Fig 2. The text is relatively small. Suggest to split this to two figures.

Fig 3. SDG 6 in (a) is not visible. Suggest making some changes.

Appendix A in SI. Very useful tables showing transparent methods and results of the interaction assessment.

Table C1 in SI: See my comments above. Which assumptions in this table address which interactions in Figures C1-C5 should be made more clear.

Page 82 in SI: This should be table D1? This table is of particular interest and useful when discussing the co-benefits of the scenarios. However, the authors did not refer to this table in the main text. Please add this. Please indicate at least in the table caption how you came up with the inputs to the table, based on the modeled results or Table C1 or both?

Response letter to comments to the manuscript "Accounting for interactions between the Sustainable Development Goals is essential for pollution control: the case of water quality in China" by Mengru Wang, Annette B.G. Janssen, Jeanne Bazin, Maryna Stokol, Lin Ma, Carolien Kroeze

22 November 2021

We appreciate the constructive comments by the two reviewers. We took advantage of all comments when revising our manuscript, which, in our opinion, largely improved our manuscript. Below, we provide details on how we incorporated the comments in the revised version.

Reviewer: 1

Comment 1:

This paper assesses the interactions between SDGs in terms of pollution control. It is critical and novel to investigate the use of SDG interactions to develop interventions and policies.

This paper is well-written and technically sound. Some major comments are provided below.

Our response: We appreciate the reviewer for acknowledging the importance of our manuscript. We also thank the reviewer for the constructive comments to improve our manuscript (see our responses to the comments below).

Comment 2:

The work has two potential contributions: 1) identified the SDGs that are relevant to nutrient pollution and assessed their interactions; and 2) developed five control scenarios and assessed their benefits in meeting the requirements. However the methodology in linking these two parts is not described. The control scenarios are reasonable and how they address the SDG interactions is also described, however, it is not clear why these scenarios are developed and if they are good control measures. It would be useful to develop a method for control measure development based on the SDG interactions. At least, clarifications should be provided on how the synergies and trade-offs between SDGs were used for developing these scenarios. For example, should priorities be placed on synergies of 'indivisible' instead of 'enabling'? how to balance promoting synergies and mitigating trade-offs if they cannot be achieved simultaneously?

Our response: The reviewer suggests clarifying why we developed the scenarios and how the identified SDG interactions were used in developing our scenarios. We appreciate this suggestion very much. We now explain more clearly the reason for developing our scenarios: "Our five alternative scenarios were developed to reduce water pollution for China while benefitting agriculture, sewage, food consumption, and climate mitigation by accounting for the interactions between the SDGs" (see track changes in the 1st paragraph of section "Methods – Scenarios" in the revised manuscript). We added clarifications on how the synergies and trade-offs between SDGs were used for developing our scenarios: "We accounted for synergies and trade-offs in developing the scenarios through the following steps. First, we made an inventory of measures that are effective in reducing nutrient pollution in Chinese water systems, based on existing scenario analyses (Ma et al., 2019; Ma et al., 2013; Stokol et al., 2017; Wang et al., 2018). Next, based on the identified SDG interactions, we identified the measures that contribute to achieving SDGs 6, 14 and SDGs 2, 11, 12 & 13 simultaneously. In other words, we tried to include in our scenarios the measures that promote synergies and avoid trade-offs between SDGs 6, 14 and SDGs 2, 11, 12 & 13." We now also provided

examples for the interactions that are addressed by measures in our scenarios (see track changes in the 1st paragraph of section “Methods – Scenarios” in the revised manuscript).

The reviewer also raises inspiring questions on how the interactions could be used to develop measures for achieving SDGs. In our study, we took the identified interactions as a ‘checklist’ to check if our scenarios for water pollution control promote synergies or mitigate trade-offs between the SDGs. There are other approaches to account for the SDG interactions in future policies. For example, the decision-makers can start from the list of the interactions, select and develop policies for the ones they would like to address. To do this effectively, important questions such as those raised by the reviewer “should priorities be placed on synergies of higher scores” and “how to balance promoting synergies and mitigating trade-offs if they cannot be achieved simultaneously” need to be answered given the regional context. We added some discussion on what other future steps/approaches can be taken to develop pollution control measures using our identified SDG interactions (see track changes in the 2nd paragraph of section “Discussion – Contribution to policies towards sustainable development” in the revised manuscript).

Comment 3:

With reference to the above comment on control measures, many other measures such as nature-based solutions or green infrastructure are generally thought to have multiple benefits. I wonder, how can these green measures be incorporated in scenarios considering the identified SDG synergies and trade-offs.

Our response: We thank the reviewer for this suggestion. In our study, we focus more on the measures that are commonly used in China to address the sources of nutrient pollution water systems. We agree that many other measures such as nature-based solutions or green infrastructure can benefit multiple SDGs too. In the revised manuscript, we added discussion on the potential benefits of other measures in achieving multiple SDGs in the perspective of water pollution control (see track changes in the 4th paragraph of section “Discussion – Model performance” in the revised manuscript).

Comment 4:

The interactions between SDGs in water pollution control are identified for China. Why should these interactions be limited to China? It seems in Tables A3 and A4 that general knowledge is applied to assess the interactions. It is not clear what specific data and knowledge about China are applied to the assessment.

Our response: The reviewer's comment includes two questions: (1) Why should the interactions between SDGs identified in this manuscript be limited to China; (2) What specific data and knowledge about China are applied to the assessment of the interactions for China. Below we addressed the two questions.

- (1) This manuscript aims to identify the interactions between SDGs from the perspective of nutrient pollution in Chinese water systems, which can serve as an example for assessing SDG interactions in other world regions. The number of interactions we identified for China (Figures 2-3 in the revised manuscript) is largely generic for other countries or regions. However, the scoring of many interactions, ranging from -3 to +3, is specific to China. These scores may be different in other countries or regions. For example, the strong trade-offs between food production (SDG 2.3) and water quality (SDG 6.3) have gained considerable

attention in the policy agendas of China. This is mainly due to the excessive fertilizer use and direct discharges of animal manure to surface waters without proper treatment, which is unique in China but less prominent in other countries today. The interactions identified in our study can thus be used as a basis and need to be contextualized (e.g., considering their spatial, governance, technology, and time-frame dependencies) when applying to other countries. In the revised manuscript, we added the discussion above in the Discussion section (see track changes in the 2nd paragraph of section “Discussion – Assessing SDG interactions” in the revised manuscript).

- (2) In the Method section of the original manuscript, we indicate that "The interactions were assessed based on existing literature (Table A1 in SI) and our expert judgment using the "Seven-Point Scale" framework from Griggs et al. (2017), who classified the interactions at 7 levels: (-3) canceling, (-2) counteracting, (-1) constraining, (0) consistent, (+1) enabling, (+2) reinforcing, (+3) indivisible". The reviewer made us realize that we need to clarify in the manuscript that we do this based on literature and our expert knowledge on Chinese water pollution. In the revised manuscript, we edited relevant text in the Method section to clarify this (see track changes in the 1st paragraph of section “Methods – Assessing synergies and trade-offs between SDG 6, 14 and other SDGs” in the revised manuscript).

Comment 5:

The use of indicators is problematic. There are clearly defined indicators for each of the targets in SDGs 6 and 14 by the UN. Why are they not used and why does China need to define its own indicators? Most importantly, the two indicators used in this paper cannot represent those SDG targets well.

Our response: The reviewer questions “*why the indicators of the SDG targets defined by UN were not used*” in the manuscript. We would like to clarify that our indicators were based on the UN-defined indicators. For SDG 6, we took an indicator based on the Chinese "Environmental Quality Standard for Surface Water" to assess the SDG target 6.3-"reducing water pollution". This indicator was defined by “China’s National Plan on Implementation of the 2030 Agenda for Sustainable Development” based on the UN-defined indicators (GPRC, 2016). For SDG 14, we took ICEP (Indicator for Coastal Eutrophication Potential) as an indicator for assessing target 14.1-"reduce marine pollution" based on the UN-defined indicator “14.1.1 (a) Index of coastal eutrophication” (IAEG, 2016). We clarified these aspects in the revised manuscript (see track changes in the section “Methods – Indicators for SDGs 6 & 14”).

To answer the question by the reviewer “*why China needs to define its own indicators*”: SDGs are global goals that need a translation in the national context for local development. China does this by translating each target of the Sustainable Development Goals (SDGs) into “action plans” considering opportunities and challenges that China faces in implementing the 2030 Agenda (GPRC, 2016). We clarified this aspect in the revised manuscript (see track changes in the 2nd paragraph of section “Methods – Indicators for SDGs 6 & 14”).

The reviewer also commented that “*the indicators used in this paper cannot represent those SDG targets well*”. We believe that our indicators represent SDG targets for the following reasons. For SDG 6.3, we took the indicator from “China’s National Plan on Implementation of the 2030 Agenda for Sustainable Development” that was developed to implement the UN’s SDG targets in the Chinese context. We thus consider the indicator we used appropriate to represent the SDG target 6.3 for

China. For SDG 14.1, we used ICEP as an indicator, following the UN-defined indicator “14.1.1 (a) Index of coastal eutrophication”. ICEP is an Indicator for Coastal Eutrophication Potential, and has been widely accepted and used in global (Garnier et al., 2010) and regional studies to assess coastal eutrophication (Romero et al., 2013; Stokal et al., 2016; Thieu et al., 2011; Wang et al., 2020). Moreover, ICEP has been recently adopted by the United Nations Environment Programme as an index to assess coastal eutrophication for SDG 14.1 (UNEP, 2021a; UNEP, 2021b). We thus consider ICEP as a representative indicator for assessing coastal eutrophication for SDG 14.1 in China. We clarified these aspects in the revised manuscript (see track changes in the section “Discussion – Indicators for SDGs 6 & 14”).

Comment 6:

How can the assumptions in the scenarios affect the analysis results? In addition to the uncertainties in the MARINA model, there are many assumptions and numbers used in each scenario. I appreciate references are provided in many places, but many numbers are very subjective and could vary significantly under different conditions. Also some numbers, such as the nutrient removal rates in SE, will also vary across different regions and river basins. Some basic sensitivity analysis is needed to understand how the benefits of pollution control are affected.

Our response: The reviewer has a good point. Our scenarios describe future sewage systems, agriculture, dietary choices, mitigation actions for N deposition and climate change. The quantitative assumptions on these aspects are based on references as indicated by the reviewer, and could indeed vary largely under different conditions. The reviewer suggests doing some basic sensitivity analysis to understand how the benefits of these scenarios are affected. We very much appreciate this suggestion and performed a sensitivity analysis. Below we describe the methods and results of the sensitivity analysis.

To investigate the impacts of uncertainties in the quantitative assumptions on the effectiveness (benefits) of the scenarios in pollution control, we performed a sensitivity analysis. This was done for 12 model inputs that are affected by the quantitative assumptions in our alternative scenarios (see Table E1 in SI). We changed these model inputs by +10% and -10% under the AG+SE+SFC+CLI scenario that includes all options for pollution control in our scenarios. This resulted in 24 model runs for 26 Chinese sub-basins. The results can be found in Tables E2-E13 in SI.

We found that the benefits of our scenarios on pollution control are, in general, not largely affected by the variations in the quantitative assumptions. The results of the sensitivity analysis show that +10% or -10% changes in 8 out of the 12 model inputs result in < 5% changes in model outputs on nutrient inputs to rivers, and river export of nutrients to seas for the sum of all sub-basins and among most individual sub-basins (Tables E2-E13 in SI). The model output for river and coastal water pollution (sum of 26 sub-basins) was somewhat more sensitive (up to 9% changes) to changes in river discharge (Tables E2-E5 in SI). Model outputs are most sensitive to changes in three model inputs: (1) population with sewage connection (2) removal fractions of nutrients during treatment in sewage systems, and (3) inputs of nutrients in animal manure to agricultural land. This is calculated for both the sum of all sub-basins (dark orange rows indicating changes in model outputs > 10% in Tables E2-E5 in SI) and individual sub-basins (number of sub-basins with changes in model outputs >10% in Tables E6-E13). This indicates the importance of improving sewage systems and animal manure management in reducing nutrient pollution in Chinese rivers.

Thus, our scenarios and sensitivity analysis show that it is technically possible to reduce water pollution in China. When formulating future policies based on our scenarios, special attention is needed for the sewage systems and manure management regarding their economic, societal, institutional and technical feasibilities to ensure the effectiveness of the policies in pollution control. We added results and discussion of this sensitivity analysis in the revised manuscript (See track changes in the 2nd and 3rd paragraphs of section “Discussion – Model performance” in the revised manuscript, and Tables E1-E13 in the Supporting Information).

Comment 7:

Lines 270-277, the example seems inappropriate as it shows the spatial trade-offs between the same goal (SDG6), while the previous statement is about the trade-offs between different SDGs.

Our response: Here we took a recent Chinese policy on banning livestock production in the so-called Non-Livestock Production Regions (NLPRs) (SCPRC, 2015; Wang et al., 2017) as an example to discuss the importance of accounting for SDG interactions in environmental policies. This policy aims to control water pollution in vulnerable water bodies by moving pig production from NLPRs to southwest and northeast provinces. With this example, we intended to show the trade-offs between SDG 6 (clean water) and SDG 15 (biodiversity on land) for China as a whole. The trade-offs occur because moving animal production reduces water pollution in South China (NLPRs), but increases air pollution and thus may cause biodiversity losses on land in Southeast and Northeast China. The reviewer has a good point that we also show the spatial trade-offs between regions in China for SDG 6 as this policy may also move the water pollution from South to Southeast and Northeast China. To make the discussion of this policy clearer, we edited our text in the revised manuscript (see track changes in the 1st paragraph of section “Discussion – Contribution to policies towards sustainable development” in the revised manuscript).

Comment 8:

It seems a section with heading 'Nutrient pollution in rivers and coastal waters' in the SI is not referenced and repeats the numbering of figures and tables.

Our response: The text in this section is indeed not being referred to in the manuscript. In the revised manuscript, we refer to this section in the main text where the results of the scenario analysis are discussed (see track changes in section “Results - Future scenarios for China accounting for co-benefits of water pollution control” in the revised manuscript). The reviewer pointed that the numbering of figures and tables are repeated in this section. This is because the results of all six scenarios are combined and shown in Figures 5, C2-C5. Therefore, the numbering may look repeating, although only three times in two pages, among the paragraphs where results of the different scenarios are discussed.

Reviewer: 2

Comment 1:

This paper addresses the important question of the interactions between the Sustainable Development Goals (SDGs), and its implications on water pollution control in China. I appreciate the great efforts by authors to analyze the >300 complex interactions between the SDG targets and develop simple scenarios as the first step to account for these interactions in policies for reducing future water pollution.

The result of this paper is original and is of crucial importance for controlling water pollution as a serious issue in China as well as the overall goal of sustainable development, which are prioritized in many Chinese policies these days. Many policies introduce guidelines/interventions for sustainable water, food and other sector's developments but temporally or spatially overlook the trade-offs between the sectors, which may cause failures of the policies. Understanding the interactions is the key to unlocking their full potential and ensuring that progress made in some areas is not made at the expense of progress in others. I've seen publications highlighting the importance of accounting for the SDG interactions in policies, or publications proposing approaches to analyze the interactions. A study like this by the authors, where they investigate the interactions at the detailed SDG target level in a specific perspective of nutrient pollution in the Chinese context, does not exist as far as I know.

This paper develops unique approaches based on an existing method to (i) identify the SDGs goals and targets that are relevant to nutrient pollution in waters of China, (ii) identify the synergies and trade-offs between the identified SDGs goals and targets, (iii) analyze the importance of considering the synergies and trade-offs in formulating environmental policies. I found the approaches used here is simple, elegant and transparent. A compliment to the detailed explanations for the interactions in the supplementary information (SI), which provide a transparent understanding of the interaction assessment and serve as a basis for replicating this study in another context.

Overall, I felt this paper contributes to our knowledge at a time when there is rising attention by both scientists and policy-makers in achieving sustainable development given the complex interactions between and within nature, society and global change. The great efforts and novel results deserve to be published after some revisions given the suggestions below.

Our response: We thank the reviewer for the compliments that our manuscript is original and is of crucial importance for controlling water pollution as a serious issue in China as well as the overall goal of sustainable development. Below we carefully addressed all comments by the reviewer.

Comment 2:

The authors analyzed the interactions between SDGs for water pollution in China. The methods and results on SDG interactions are relatively clear. A question to the discussion of these results: are these interactions beneficial for other countries which also have problems with nutrient pollution? To what extent are these results specific for China? Or in other words, if others take these interactions for another country/region, what do they first need to check before applying them?

Our response: Reviewer 1 had a similar comment. We realized that we did not discuss enough how our results are useful to other countries experiencing nutrient pollution in water. We now elaborate

our discussions in the revised manuscript (see track changes in the section “Discussion – Assessing SDG interactions”). Please see also our response to Comment 4 by Reviewer 1.

Comment 3:

Another question to interaction assessment. What are the uncertainties related to this assessment? The seven-point approach involved subjectivity in any case. How does this influence the results/conclusions? Suggest adding some discussion on this.

Our response: The reviewer points out that the “Seven-Point Scale” approach involves subjectivity during the assessment and suggests adding some discussion on how this influences the results/conclusions of our study. As suggested by the reviewer, we elaborated our discussion on the strengths and weaknesses of the “Seven-Point Scale” approach: “We realize that the scoring of the interactions is inevitable to some extent subjective because it is in part based on interpretation. We combined scientific evidence and expert judgment to assess the interactions. When another team would conduct the same process, the resulting scores may be different, depending on their interpretations of the scores and the growing scientific evidence on the interactions. We consider this, however, a strength, as long as the assessment is transparent, as done in this study (see scores and explanations in Tables A3-A4). This allows other scientists and stakeholders to take our assessment of interactions and further improve this assessment based on new scientific literature and their own expertise.” (see track changes in the 1st paragraph of Section “Discussion – Assessing SDG interaction” in the revised manuscript).

Comment 4:

The authors analyzed the co-benefits of water pollution control that benefits from synergies and avoids trade-offs based on scenario analysis. They indicate the interactions that can be addressed by their scenarios in the SI. To me, it is not very clear which assumptions/options in the scenarios in Table C1 address which the interactions in Figures C1-C5. Suggest making this clear.

Our response: The reviewer provides a very good suggestion to indicate clearly which assumptions in our scenarios in Table C1 address which interactions in Figure C1-C5. Following this suggestion, we added a table where we list the interactions that may be addressed by each specific assumption in our scenarios (see Table C2 in the revised Supporting Information with track changes).

Comment 5:

The co-benefits of water pollution control by promoting synergies and avoiding trade-offs on other SDGs is not assessed by the model but based on the assumptions in the scenario as I understood. This is fine as the first try-out to account for the interactions. I know such assessment is complex and will need to integrated models from different disciplines, which is challenging and is, anyway, not the main goal of this paper. However, a more elaborated discussion on how next steps/effort, especially using integrated modeling (e.g., combining food, water, hydrological, economic and etc. models), can be taken to assess the effectiveness of accounting for interactions in future policies is needed.

Our response: The reviewer has a very good point that further effort is needed to develop and apply an integrated modeling approach to better understand the effectiveness of accounting for interactions in future policies. As suggested by the reviewer, we elaborated our discussion on

accounting for the SDG interactions in policies by adding the following text: “In our study, the co-benefits of water pollution control are, however, assessed based on the assumptions in the scenarios and not by the model (see Methods). This is because integrated modeling frameworks that combine models from different disciplines (e.g., food, water, hydrological, economic) are scarce. Such modeling frameworks are strongly needed as they could contribute to (1) assessing multiple SDGs under consistent climate and socio-economic conditions, (2) quantifying the SDG interactions, and (3) supporting decision-makers by modeling the effectiveness of accounting for the SDG interactions in their policies. Thus, to better account for the interactions in future policies, further research is needed on developing and applying system (e.g., integrated modeling) approaches to analyze the SDG interactions.” (see track changes in the 2nd paragraph of section “Discussion - Contribution to policies towards sustainable development” in the revised manuscript).

Comment 6:

In the discussion, the authors pointed out an important issue: there is a general lack of quantitative indicators to monitor the progress of SDG6. This is a very good point. The authors discussed the possibilities to develop indicators based on the water quality standards in other countries, which may be challenging in my view as the standards differ largely among countries. There are existing indicators to assess water scarcity based on both quality and quantity (<https://www.nature.com/articles/ngeo3047>). Why not include them?

Our response: We thank the reviewer for this very good suggestion. The water scarcity indicator is based on both water quality and quantity, and can be useful to monitor the progress of SDG6. We added discussion on this indicator in our discussion section (see track changes in the 2nd paragraph of section “Discussion – Indicators for SDGs 6 & 14” in the revised manuscript).

Comment 7:

Line 60. Please refer to the SI or Method where this seven-point scale approach is described. The report is rather long for authors to find their description of the approach.

Our response: As suggested by the reviewer, we edited the text to refer to the SI and Method where the seven-point scale approach is described in the revised manuscript (see track changes in the last paragraph of section “Introduction” in the revised manuscript).

Comment 8:

Line 87. "nutrient pollution in waters" -> "water pollution by nutrients"

Our response: We edited the sentence as suggested by the reviewer (see track changes in the 3rd paragraph of section “Results - Identifying Interactions between SDGs” in the revised manuscript).

Comment 9:

Line 118-120. What negative side-effects? Could you provide some examples to make it more clear?

Our response: We added examples of the negative side-effects in the revised manuscript (see track changes in the last paragraph of section “Results - Identifying Interactions between SDGs” in the revised manuscript).

Comment 10:

Line 133. Also consider the ongoing pandemic?

Our response: We agree with the reviewer that the ongoing pandemic also challenges achieving all SDGs in 2030. We added text on this in the revised manuscript (see track changes in the 1st paragraph of section “Results - Future scenarios for China accounting for co-benefits of water pollution control” in the revised manuscript).

Comment 11:

Line 159. Reference needed for China's water quality standard.

Our response: We added the reference for China's water quality standard in the revised manuscript (see the 3rd paragraph of section “Results - Future scenarios for China accounting for co-benefits of water pollution control” in the revised manuscript).

Comment 12:

Line 228. Very good that the authors show the quantitative ratings for the model assessment

Our response: We thank the reviewer for this compliment.

Comment 13:

Line 350-354. See my comments above, which assumptions address which interactions should be made more explicit, at least in the SI.

Our response: We added a table to show the interactions that are addressed by each specific assumption in our scenarios (see Table C2 in the revised Supporting Information with track changes). Please see also our response to Comment 4 by this reviewer.

Comment 14:

Fig 2. The text is relatively small. Suggest to split this to two figures.

Our response: We agree with the reviewer and split the figure into two figures (see updated Figures 2 and 3 in the revised manuscript).

Comment 15:

Fig 3. SDG 6 in (a) is not visible. Suggest making some changes.

Our response: We modified the figure to make SDG 6 visible (see updated Figure 4 in the revised manuscript).

Comment 16:

Appendix A in SI. Very useful tables showing transparent methods and results of the interaction assessment.

Our response: We thank the reviewer for this compliment.

Comment 17:

Table C1 in SI: See my comments above. Which assumptions in this table address which interactions in Figures C1-C5 should be made more clear.

Our response: We added a table to show the interactions that are addressed by each specific assumption in our scenarios (see Table C2 in the revised Supporting Information with track changes). Please see also our response to Comment 4 by this reviewer.

Comment 18:

Page 82 in SI: This should be table D1? This table is of particular interest and useful when discussing the co-benefits of the scenarios. However, the authors did not refer to this table in the main text. Please add this. Please indicate at least in the table caption how you came up with the inputs to the table, based on the modeled results or Table C1 or both?

Our response: We thank the reviewer for pointing out the mistake in the table numbering. The table should be Table C3. We corrected this in the revised Supporting Information.

Used references in this response letter

- Garnier J., Beusen A., Thieu V., Billen G., Bouwman L. N: P: Si nutrient export ratios and ecological consequences in coastal seas evaluated by the ICEP approach. *Global Biogeochemical Cycles* 2010; 24.
- GPRC. China's National Plan on Implementation of the 2030 Agenda for Sustainable Development. In: China GotPsRo, editor, 2016.
- Griggs D., Nilsson M., Stevance A., McCollum D. A guide to SDG interactions: from science to implementation: International Council for Science, Paris, 2017.
- IAEG U. Final list of proposed Sustainable Development Goal indicators. Report of the Inter-Agency and Expert Group on Sustainable Development Goal Indicators (E/CN. 3/2016/2/Rev. 1), 2016.
- Ma L., Bai Z., Ma W., Guo M., Jiang R., Liu J., Oenema O., Velthof G.L., Whitmore A.P., Crawford J., Dobermann A., Schwoob M., Zhang F. Exploring Future Food Provision Scenarios for China. *Environmental Science & Technology* 2019; 53: 1385-1393.
- Ma L., Wang F., Zhang W., Ma W., Velthof G., Qin W., Oenema O., Zhang F. Environmental assessment of management options for nutrient flows in the food chain in China. *Environmental science & technology* 2013; 47: 7260-7268.
- Romero E., Garnier J., Lassaletta L., Billen G., Le Gendre R., Riou P., Cugier P. Large-scale patterns of river inputs in southwestern Europe: seasonal and interannual variations and potential eutrophication effects at the coastal zone. *Biogeochemistry* 2013; 113: 481-505.
- SCPRC. Action Plan for Prevention and Control of Water Pollution (In Chinese). In: China TSCTPsRo, editor, Beijing, China, 2015.
- Strokal M., Kroeze C., Wang M., Bai Z., Ma L. The MARINA model (Model to Assess River Inputs of Nutrients to seAs): Model description and results for China. *Science of The Total Environment* 2016; 562: 869-888.
- Strokal M., Kroeze C., Wang M., Ma L. Reducing future river export of nutrients to coastal waters of China in optimistic scenarios. *Science of The Total Environment* 2017; 579: 517-528.
- Thieu V., Billen G., Garnier J., Benoît M. Nitrogen cycling in a hypothetical scenario of generalised organic agriculture in the Seine, Somme and Scheldt watersheds. *Regional Environmental Change* 2011; 11: 359-370.
- UNEP. 14.1.1(a) Index of coastal eutrophication; and (b) plastic debris density. 2021, 2021a.
- UNEP. Understanding the State of the Ocean: A Global Manual on Measuring SDG 14.1.1, SDG 14.2.1 and SDG 14.5.1, Nairobi, 2021b.
- Wang M., Kroeze C., Strokal M., van Vliet M.T.H., Ma L. Global change can make coastal eutrophication control in China more difficult. *Earth's Future* 2020; n/a: e2019EF001280.
- Wang M., Ma L., Strokal M., Chu Y., Kroeze C. Exploring nutrient management options to increase nitrogen and phosphorus use efficiencies in food production of China. *Agricultural Systems* 2018; 163: 58-72.
- Wang Y., Wang X., Li M., Wu L., Zhang N., Xu F. Status Quo of and Problems in Delimiting of Livestock/Poultry Breeding Forbidden Zones in China (in Chinese). *Journal of Ecology and Rural Environment* 2017; 33: 615-621.

Reviewer comments, second round review -

Reviewer #1 (Remarks to the Author):

I appreciate the effort from the authors in addressing my comments, in particular adding new sensitivity analysis and discussion. All my comments are addressed satisfactorily and I have no further comments.

Reviewer #2 (Remarks to the Author):

My comments have been well addressed. It could be published now.

Response letter to comments to the manuscript "Accounting for interactions between Sustainable Development Goals is essential for water pollution control in China" by Mengru Wang, Annette B.G. Janssen, Jeanne Bazin, Maryna Stokal, Lin Ma, Carolien Kroeze

21 December 2021

We appreciate the positive comments by the two reviewers.

Reviewer: 1

Comment 1:

I appreciate the effort from the authors in addressing my comments, in particular adding new sensitivity analysis and discussion. All my comments are addressed satisfactorily and I have no further comments.

Our response: We appreciate the reviewer for acknowledging our efforts in addressing his/her comments. We are happy to hear that all comments were addressed satisfactorily.

Reviewer: 2

Comment 1:

My comments have been well addressed. It could be published now.

Our response: We are happy that the reviewer's comments have been well addressed, and the reviewer agrees that our manuscript could be published now.